# Actinotrichia-independent developmental mechanisms of spiny rays facilitate the morphological diversification of Acanthomorpha fish fins

Kazuhide Miyamoto [1] ✉, Junpei Kuroda[2,3], Satomi Kamimura[4], Yasuyuki Sasano[5], Gembu Abe [6], Satoshi Ansai[7], Noriko Funayama[8], Masahiro Uesaka [1] & Koji Tamura[1]

Skeletal forms in vertebrates have been regarded as good models of morphological diversification. Fish fins show great diversity in form, with their supporting skeletal structure being classified into soft rays and spiny rays. In fish evolution, spiny-ray morphologies are known to be sometimes extremely modified; however, it remains unknown how the developmental mechanisms of spiny rays have contributed to their morphological diversification. By using the rainbowfish *Melanotaenia praecox* for examination of the extracellular matrix (ECM) and cell dynamics of spiny-ray development, we demonstrate that spiny-ray development is independent of the actinotrichia (needle-shaped collagen polymers at the tip of fins), which are known as an important ECM in soft-ray morphogenesis. Furthermore, we found that in the thorny spiny ray of the filefish *Stephanolepis cirrhifer*, the lateral protrusions are associated with BMP-positive osteoblast condensation, as in the spiny-ray tips in *M. praecox* and *S. cirrhifer*. Taken together, our findings reveal that osteoblast distribution and signaling-molecule intensity would contribute to spiny-ray modification. In comparison to soft ray development, the independence from actinotrichia in spiny rays would facilitate growth direction change, leading to their morphological diversification. This suggests that variation in cell distribution and ECM usage may be important contributors to morphological diversification, not only in Acanthomorpha, but also in other animal taxa.

Understanding the fundamental mechanisms of morphological diversification is an ultimate goal of evolutionary biology. Skeletal forms in vertebrates are highly divergent[1–3], and the molecular and genetic mechanisms of skeletal development have been extensively studied. Although skeletal forms serve as a good model of morphological diversification, the contribution of underlying developmental mechanisms to morphological diversification remains unknown.

Fish fin bones are among the most diversified organs among vertebrate skeletal forms and have acquired various functions and facilitate adaptations to various habitats[4,5]. In teleosts, the fins are commonly supported by bone structures called fin rays[1,3,6–9], which are further classified into soft rays and spiny rays[8,10–12] (Fig. 1a). Soft rays are flexible, segmented, and typically branched, whereas spiny rays are stiff, unsegmented, and terminate in an acute point[5,11,13–15]. Spiny rays

have evolved independently in several lineages, such as the order Siluriformes and the clade Acanthomorpha[11,15–17] (Fig. 1b). The Acanthomorpha is one of the largest lineages of vertebrates[18–21], and their spiny rays are called "true spines"[5,7,11,18,22]. Interestingly, true spines are known to be extremely modified to form novel structures that do not work as fin components but as species-specific apparatuses. For example, pectinated lamellae, which form the sucking disc of remoras[23,24], and illicium, which is the fishing apparatus of goosefishes and anglerfishes[8,25], originated from the spiny rays of ancestral fish. Spiny rays could be regarded as an easily diversified skeletal feature,

and have long fascinated scientists[26]. In contrast, soft rays sometimes evolved to be elongated in several lineages (e.g., in the pectoral fins of flying fish, and the dorsal fin of sailfish), but tend not to have been modified into non-rod-like shapes in their evolution[5]. Thus, the spiny-ray morphology is considered highly diversified[5]. Presently, little is known about factors that lead to this difference in the morphological diversity between spiny and soft rays. Developmental mechanisms of spiny rays have been largely unknown, although the development and regeneration process of soft rays have been well studied[27–34]. While it is possible that the differences in the developmental mechanisms

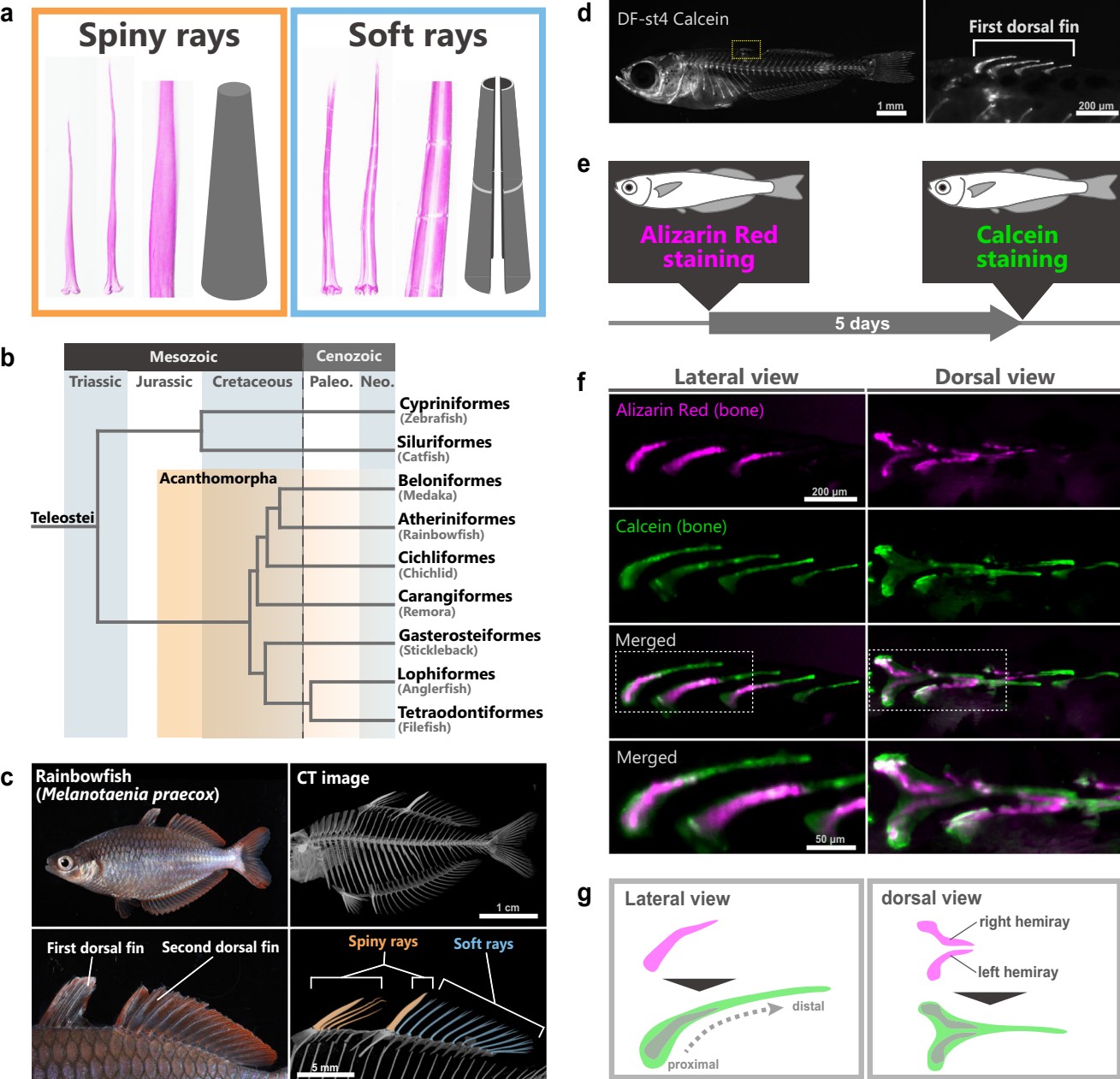

**Fig. 1 | Morphology and development of spiny rays in *Melanotaenia praecox*.** **a** Specimens stained with alizarin red and schematic illustration of spiny rays (left) and soft rays (right) in an Acanthomorpha fish. The spiny- and soft-ray samples stained with alizarin red are the dorsal fins of *Melanotaenia praecox*. **b** Simplified phylogenetic tree of the Teleostei (modified from Figs. 1 and 2 in Ghezelayagh et al.[19]. and Fig. 2 in Hughes et al.[73]). **c** Lateral views of an adult *Melanotaenia praecox*; the lower panels are magnified images of the dorsal-fin regions shown in the upper panels. (CT image, adult, *n* = 3) **d** Calcein-staining image of a DF-st4[12] larva (*n* = 10), and magnified image of the first-dorsal fin area marked by the yellow dashed box. **e** Schematic images of the two-color staining procedure. **f** Bone development of spiny rays in the first dorsal fin revealed by double staining: the fluorescence of spiny-ray bones visualized by each probe, as revealed by alizarin red (magenta) and calcein (green), and merging of the two stains (*n* = 10). Magnified image in the lower panels marked by the white dashed box. **g** Schematic illustration of the growth of the spiny-ray bone.

between soft- and spiny-ray morphologies might be a candidate cause, we have not yet understood the contribution of spiny-ray developmental mechanisms to their morphological diversification.

Therefore, to uncover the developmental mechanisms of spiny rays, we sought to examine their morphogenesis at the cellular and molecular levels using laboratory animals. However, a critical problem was that popular model fish, such as zebrafish and medaka, could not be used for spiny-ray research. This is because zebrafish are not an Acanthomorpha and lack this structure[11] and medaka have degenerated their spiny rays in the dorsal fin and have only one small spiny ray in their anal fin[35]. Consequently, for this study, we chose the dwarf neon rainbowfish *Melanotaenia praecox*, a small freshwater fish in the order Atheriniformes (Fig. 1c), which has spiny rays in the dorsal, anal, pectoral, and pelvic fins[12], as a new model fish. Furthermore, we have already established a staging system for its postembryonic development based on fin development[12], as well as a genome editing system for *M. praecox*[36].

Here, we examined the extracellular matrix (ECM) and cell dynamics of spiny-ray development in *M. praecox* and compared them with those of soft-ray development. Our data reveal several differences in developmental mechanisms between spiny and soft rays, such as their dependence on or independence from the actinotrichia, which are known as an important ECM structure in zebrafish fin-ray growth and regeneration[31,32,37,38]. To gain insight into spiny-ray modification in Acanthomorpha evolution, we also investigated the dynamics of premature osteoblasts and bone morphogenetic protein (BMP) signaling in these cells in the filefish *Stephanolepis cirrhifer*, which has extremely modified spiny rays on its back, as well as in *M. praecox*. Based on our observations, we propose that the developmental mechanisms of spiny rays may have facilitated their morphological diversification.

## Results

### Growth dynamics of the spiny ray

Soft rays grow by adding bone matrix to the distal tip[34]. However, the growth patterns of spiny rays are largely unknown. To identify differences in morphogenesis between soft and spiny rays, we first visualized their developmental changes using two different staining methods[12]. We first used alizarin red for staining calcified bone in *M. praecox* dorsal-fin stage 4 (DF-st4) larvae[12], of which some spiny rays were observed in their first dorsal fin (Fig. 1d); we then re-stained the bones with calcein, a green-fluorescent molecule, 5 days after the first staining with alizarin red (Fig. 1e). Hence, this two-color staining of the larvae (DF-st4 larvae, $n = 10$) showed the morphology of their spiny rays at two time points (Fig. 1e–g). By comparing the morphology at different time points, we found that in each spiny ray, the calcein-stained area extended distally from the alizarin red/calcein double-positive area (Fig. 1f, g). This result suggests that the spiny rays mainly grow by adding bone matrix to their distal tips. Furthermore, the spiny rays also grow radially by adding bone matrix, since the calcein single-positive area surrounded the alizarin red/calcein double-positive area. The dorsal view showed that the spiny rays were composed of left and right elements at the time of alizarin red staining, and a thick single bone element at the time of calcein staining (Fig. 1f, g). The right and left elements seem fused by the radial growth of each element, which is consistent with previous morphological descriptions of the spiny-ray character[5,15].

### Actinotrichia do not play a crucial role in spiny-ray growth

Bone staining of *M. praecox* larvae with two different-colored staining molecules revealed that the spiny rays grow by adding bone matrix to their distal tips, which is similar to the growth process observed in soft-ray growth[32,34,39]. Thus, spiny rays may share some growth mechanisms with soft rays. In soft rays, actinotrichia (needle-shaped collagen polymers) are bundled at their distal tips and serve as scaffolds for mesenchymal cell migration during bone formation in fins[29,31–33,37–42].

To understand the function of spiny-ray development, we focused on the actinotrichia distribution in *M. praecox* fin fold and fins by using DAFFM (diaminofluorescein-FM) staining[32,43,44] at the early stages of soft- and spiny-ray growth (Fig. 2a). DAFFM is a small fluorescent molecule that has been broadly used for NO detection, and recent research has shown that DAFFM can bind to collagenous structures like actinotrichia[32,43–46]. At the tips of soft rays in the second dorsal fin of larvae and juveniles (DF-st3 larvae, $n = 3$; DF-st4 larvae, $n = 4$; juveniles, $n = 4$), we found that actinotrichia were arranged and bundled during soft-ray development (Fig. 2a). Before the emergence of spiny rays (i.e., in DF-st3 larvae), relatively small and less developed actinotrichia, compared with those at the tips of the soft rays, were observed in the median fin fold, including regions that later give rise to the spiny rays. At the initial stage of spiny-ray development (at DF-st4), fine actinotrichia were aligned in parallel with spiny-ray primordia, but these were not distributed at the tip of each spiny ray, as seen in the soft rays. At the juvenile stage, actinotrichia were not detected around the spiny rays (Fig. 2a). DAFFM staining visualizes collagenous structures, including actinotrichia, bone collagen, and tendon of the segments in the soft rays[32,44]. Therefore, the bright signals in Fig. 2a correspond mainly to actinotrichia, bone collagen, and tendon of the segments in the soft rays.

We further confirmed the production of actinotrichia around spiny rays in *M. praecox* by determining the expression of the gene *actinodin1* (*and1*) and *actinodin2* (*and2*), which encodes proteins known to be essential structural components of actinotrichia[28,37,38]. At DF-st4 (*and1*, DF-st4 larvae, $n = 6$; *and2*, DF-st4 larvae, $n = 4$) and the juvenile stage (*and1*, juvenile, $n = 4$; *and2*, juvenile, $n = 4$), *and1* and *and2* were strongly expressed at the tip of each soft ray in the second dorsal fin (Supplementary Fig. 1). However, at both stages, there was no *and1* expression and no or weak *and2* expression around spiny rays in the first and second dorsal fins (Supplementary Fig. 1). These results indicate that, at the tips of spiny rays, actinotrichia are neither actively formed nor maintained.

It is reported that actinotrichia in the fin folds were absent in zebrafish *and1* and *and2* morphants[38]. Therefore, we investigated this in *M. praecox* by first making a double knockout (KO) of genes *and1/and2* using the CRISPR/Cas system (Supplementary Figs. 9 and 10). We isolated some types of mutant fish with several deletions in the *and1/and2* genes, each of which causes a frame-shift mutation and is expected to encode a non-functional truncated protein (Supplementary Figs. 2 and 3). Next, to assess the role of the actinotrichia in spiny-ray growth, we generated mutant *M. praecox* with the genotype *and1⁻/⁻/and2⁺/⁻* (DF-st4 or juveniles with an SL < 1 cm, $n = 6$) (Supplementary Figs. 2 and 3), for comparison of the actinotrichia condition and soft/spiny-ray morphologies in wild-type fish (DF-st4 or juveniles with a SL < 1 cm, $n = 8$) (Fig. 2b) and the mutant fish (Fig. 2c). In mutant fish images of the DAFFM and alizarin red staining, we observed that the actinotrichia were not radially arranged as in wild-type fish (Fig. 2b), and that the soft rays were distorted and curled in the mutant fish (white arrowheads in Fig. 2c and Supplementary Fig. 4). We observed that the intervals of the segments of the soft rays were uncertain (yellow arrowheads in Fig. 2c and Supplementary Fig. 2). The restriction of fin ray growth may result in the shortening of each soft ray. The curled phenotypes were not always observed in the mutant fish, with this phenomenon being caused by the edge of the caudal part of the fin fold failing to grow. Rather, the mutants had a normal, straight spiny-ray bone morphology similar to that of the wild-type fish (Fig. 2b, c). Furthermore, we generated *M. praecox* mutants with the genotype *and1⁺/⁻*; *and2⁻/⁻* (DF-st4 or juveniles with SL < 1 cm, $n = 2$) (Supplementary Figs. 5 and 10). In these mutants, we observed the actinotrichia were radially arranged and that the soft rays were slightly distorted (white arrowhead in Supplementary Fig. 5). Although weak *and2* expression was detected around the spiny rays in some in situ hybridization specimens (Supplementary Fig. 1), the spiny rays of the

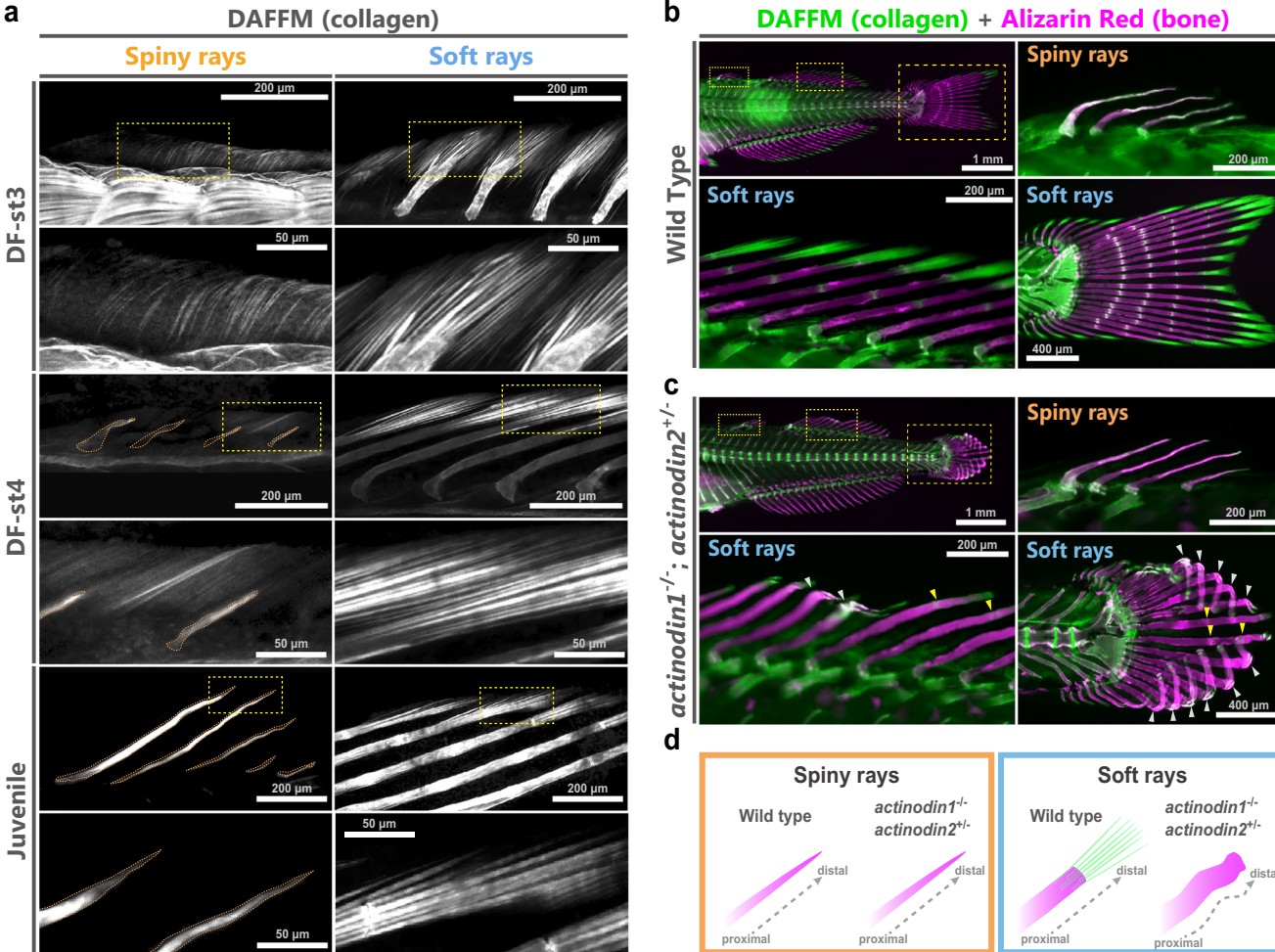

**Fig. 2 | Role of actinotrichia in spiny- and soft-ray morphogenesis in *Melanotaenia praecox*. a** Distribution of the actinotrichia at the tips of spiny rays (left) and soft rays (right) at three stages of fin deveolpent[12] in fixed samples (DF-st3 larvae, *n* = 3; DF-st4 larvae, *n* = 4; juveniles, *n* = 4); the lower pair of panels of each stage are magnified images of the yellow dashed box in the upper panels. Spiny-ray bones are outlined in orange. Actinotrichia distribution and spiny- and soft-ray bone morphology in wild-type fish (**b**) (*n* = 8) and in the *actinodin1⁻/⁻/actinodin2⁺/⁻* knockout fish (**c**) (*n* = 6). All panels except the top-left panel represent magnified views of the area enclosed by the yellow dashed box in the top-left panel. White arrowheads indicate abnormal bending of the soft rays in the knockout fish. Yellow arrowheads show aberrantly positioned segmental structures in the soft rays of the knockout fish. Actinotrichia were labeled with DAFFM DA (green), and the spiny- and soft-ray bones were labeled with alizarin red (magenta). **d** Schematic illustration of the spiny- and soft-ray morphology in fish with a loss of actinotrichia.

*and1⁺/⁻; and2⁻/⁻* mutants were not affected by the *and2* knockout (Supplementary Fig. 5). Regarding the function of the actinotrichia, our observations strongly suggest that actinotrichia do not play a role in the growth and morphogenesis of spiny rays (Fig. 2d). The actinotrichia seen early in the first dorsal fin in DF-st4 larvae could be what remains from supporting the median fin fold before that stage, as seen in the larval median fin fold of zebrafish, where actinotrichia are present and surmised to be essential for fin-fold development and maintenance[32,38,40,47]. Considering this together, the presence of actinotrichia around the spiny rays in the DF-st4 larvae would be consistent with the condition in the mutant fish. Therefore, we next asked how spiny rays could achieve distal growth independently of the actinotrichia.

**Mesenchymal cell condensation at the spiny-ray tips**
To explore the growth mechanisms of spiny rays, we observed the cellular conditions at the tips of spiny rays in comparison with those of soft rays. First, we examined the 3D cell distribution at the tips of spiny rays (DF-st4 larvae, *n* = 3) and soft rays (DF-st4 larvae, *n* = 3) by staining the nucleus and collagen structures simultaneously. DAR-4M is a small

fluorescent molecule similar to DAF-FM that is widely used for NO detection; however, the two molecules exhibit different fluorescence emission wavelengths[32,44]. Recent studies have shown that DAR-4M can bind to collagenous structures such as actinotrichia[32,46]. We found that mesenchymal cells are multilayered at the tips of spiny rays (Fig. 3a-left), whereas such condensed mesenchymal cells were not detected at the tips of soft rays (Fig. 3a-right)".

We next observed the detailed conditions of the mesenchymal cells and ECM. Transmission electron microscopy (TEM) images of transverse sections of soft rays (DF-st4 larvae, *n* = 3) revealed several thick actinotrichia (magenta coloration) aligned between the two hemirays (light blue coloration), with a few mesenchymal cells in the space between the hemirays, as previously reported in zebrafish[48] (Supplementary Fig. 6). Furthermore, the hemirays closely adjoined the basement membrane, and no mesenchymal cells were distributed between the hemirays and the basement membrane. TEM images of transverse sections of spiny rays (DF-st4 larvae, *n* = 3) (Fig. 3b) showed very few or no actinotrichia, consistent with our staining observations indicating the absence of actinotrichia (Fig. 2a). Mesenchymal cells (green coloration) were highly condensed surrounding the spiny ray

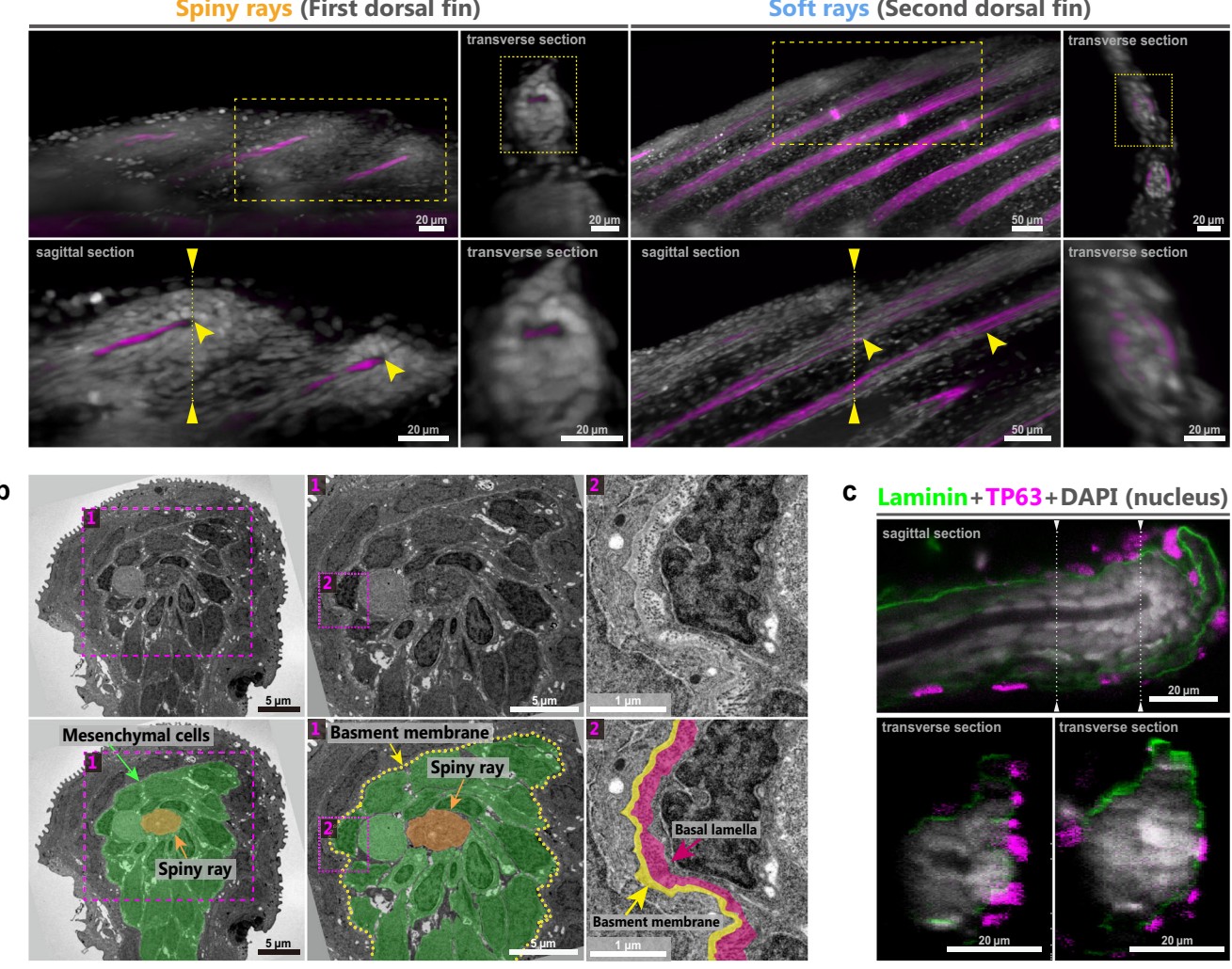

**Fig. 3 | The cell and extracellular matrix (ECM) conditions at the tips of spiny rays in *Melanotaenia praecox*. a** Cell, collagen, and bone conditions of the dorsal fin in a larva at DFst-4 ($n = 3$). The upper left panels are 3D light-sheet images of the dorsal fins, and the other panels are virtual sections of the 3D images. The tips of spiny rays and soft rays are indicated by yellow arrowheads. The location of the virtual sagittal section of the spiny and soft rays shown in the lower left panels is indicated by the yellow dashed box in the upper left panels. The location of the virtual transverse section of the spiny and soft rays shown in the right panels of each fin is indicated by the dashed line in the lower left panels. The lower right panels of each fin are magnified images of the yellow dashed box in the upper right panels. **b** Ultrastructural analysis of the spiny rays in DF-st4 larvae by TEM ($n = 3$). First-dorsal-fin sections showing the structural organization of the spiny-ray tip. The lower panels show the TEM images labeled for mesenchymal cells (green), a spiny ray (orange), basement membrane (yellow) and basal lamella (hot pink). The numbers of each panel (#1, #2) correspond to the numbers in magenta dashed boxes. **c** Distribution of laminin (green) and TP63-positive cells (magenta) in the spiny rays of a DF-st4 larva, counter-stained with DAPI (white) ($n = 9$). Images in panel **c** are virtual sections of the same sample by confocal microscopy. The locations of the virtual transverse section shown in the lower panels are indicated by the dashed line in the upper panel.

(orange coloration), and the layer composed of basement membrane (yellow coloration) and basal lamella[49,50] (hot pink coloration) surrounded this cell condensation (Fig. 3b). Furthermore, the basement membrane around spiny rays was thicker than the basement membrane seen next to soft rays (Supplementary Fig. 6). We then focused on the distribution of laminin proteins, a major component of the basement membrane[51,52]. Three-dimensional reconstructions of immunohistochemistry analysis at the tip of a spiny ray showed that the signal for laminin specifically accumulated beneath the TP63-positive epidermal basal cells (DF-st4 larvae, $n = 9$). The distribution of laminin strongly suggests that the laminin layer surrounding the mesenchymal condensation at the tips of spiny rays in TEM images (Fig. 3b) is the basement membrane, with basal lamella forming a cap-like structure on the spiny-ray tips (Fig. 3c). This structure was not

detected at the tips of soft rays (Supplementary Fig. 6), nor has it been reported in developing and regenerating soft rays where the actinotrichia are distributed at the tip.

The differences in the ECM and cellular conditions of developing spiny and soft rays are shown in Fig. 4 and demonstrate that the developmental mechanism of the spiny rays is independent of actinotrichia, which are a vital factor for the straight growth of soft rays. Rather, it is possible that the condensed mesenchymal cells at the tip of developing spiny rays may play a pivotal role in spiny-ray bone formation.

## BMP signaling for proper osteogenesis of spiny rays

Given that mesenchymal cells surround the tips of spiny rays, we considered that they may contribute to the morphogenesis of the

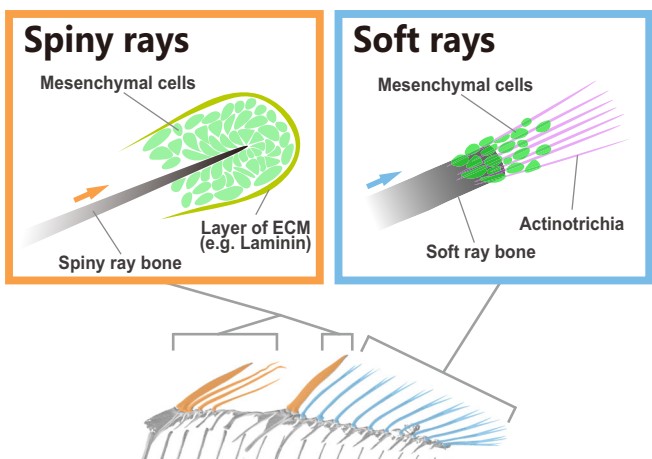

**Fig. 4 | Schematic illustration showing differences between spiny- and soft-ray development in *Melanotaenia praecox*.** In spiny-ray development, mesenchymal cells (green) condense at the tips of spiny-ray bones, and the layer of extracellular matrix (ECM) (yellow) surrounds mesenchymal cells. In soft-ray development, the actinotrichia (magenta) are distributed at the tips of soft-ray bones and serve as scaffolds for mesenchymal cells (green). Micro-CT scan images were used to schematically illustrate the fin bones.

spiny ray through their osteogenesis. Therefore, we examined the osteoblast distribution and osteogenic signaling pathways in these cells. The upstream osteogenic gene *runx2*[53] is known to be expressed in pre-osteoblasts at the distal tip of regenerative blastema in the caudal fin of zebrafish[54,55]. Moreover, BMP signaling promotes osteoblast differentiation[56]. Accordingly, we next observed the distribution of premature osteoblasts and BMP signaling in the mesenchymal cell condensation around the tip of the spiny ray in *M. praecox* larvae using immunohistochemistry (DF-st4 larvae, *n* = 9) (Fig. 5). Runx2 signals were specifically detected in the cell condensations of the distal tips of the spiny rays, and active Smad1, 5, and 9 (pSmad 1/5/9) signals were detected in some cells of the Runx2-positive cells (Fig. 5a). This suggests that the cell condensation at the distal tips of spiny rays was composed of pre-osteoblasts, and that BMP signaling may contribute to the osteogenesis of spiny rays.

To verify the role of BMP signaling in spiny-ray osteogenesis, we administrated a BMP-receptor inhibitor, DMH1. In DMH1-treated fish (DF-st4 larvae, *n* = 8), pSmad 1/5/9 signals were lost in the cell condensation, although they were clearly detected in control fish (DF-st4 larvae, *n* = 7) (Fig. 5b), thereby indicating that DMH1 properly inhibited BMP signaling at the spiny-ray tip (Fig. 5b). In the immunohistochemistry images for pSmad1/5/9 (leftmost panels of Fig. 5b), several punctate signals are visible in both control and DMH1-treated fish. In previous studies using the same antibody, pSmad1/5/9 immunoreactivity was prominent in nuclei[57,58]. Therefore, the punctate signals observed in our images may represent non-specific staining. Surprisingly, the tips of spiny rays in the DMH1-treated fish were largely deformed and much thicker than in the control fish, where several cells were buried in the deformed spiny-ray bone in some of the treated specimens (DF-st4 larvae, *n* = 5/8) (blue arrowheads in Fig. 5b), which we did not see in the spiny rays of the controls (DF-st4 larvae, *n* = 0/7) (Fig. 5b). Despite BMP signaling inhibition by DMH1, the spiny rays retained their characteristic unsegmented and rigid morphology. No actinotrichia-like collagenous structures or segmentation were observed at the tips of the spiny rays in treated specimens (Fig. 5b). Taken together, these results suggest that premature osteoblasts condense at the spiny-ray tip, which is the most readily growing part of this bone, and that BMP signaling in these cells is required for proper spiny-ray morphogenesis.

## Density of mesenchymal cells in modified spiny rays

Developmental mechanisms of spiny rays in *M. praecox* imply that the distribution of premature osteoblasts and working BMP signaling should be a key factor for regulating the spiny ray morphology. Based on these findings, we hypothesized that alternation of the osteoblast distribution and BMP signaling in these cells have resulted in morphologically modified spiny rays in the course of Acanthomorpha fish diversification (Fig. 6). To test the hypothesis, we observed the premature osteoblast cell distribution of spiny rays in the filefish *Stephanolepis cirrhifer* (Fig. 6a), a member of the order Tetraodontiformes (Fig. 1b). Larva of *S. cirrhifer* has a thorny dorsal spine, which is a modified spiny ray (Fig. 6a, b, CT-image, 0.98 cm and 1.3 cm SL, *n* = 2). Runx2-positive premature osteoblasts were condensed at the tip of the dorsal spine in *S. cirrhifer* (Fig. 6c and Supplementary Fig. 7, 1.2–1.35 cm SL, *n* = 3), and some of these cells had pSmad1/5/9 signals (yellow arrowhead in Fig. 6c and Supplementary Fig. 7), as seen in the spiny-ray tips in *M. praecox* larvae (Fig. 5a). Furthermore, in protrusions (thorn primordia) that branch from the stem of a dorsal spine, premature osteoblasts (Fig. 6c and Supplementary Fig. 7) specifically form a thin layer, and pSmad1/5/9 positive signals (yellow arrowhead in Fig. 6c and Supplementary Fig. 7) were detected in these cells. The cellular conditions of the *S. cirrhifer* suggest that the mechanisms adding bone at the tip of the stem of that spiny ray also act on the tip of the protrusions in the thorny dorsal spine, consistent with our hypothesis.

## Discussion

Our study confirms that some features of the morphogenetic mechanisms underlying spiny-ray development in acanthomorph fish differ from those of soft-ray development. Our data suggest that the actinotrichia do not play an important role in spiny-ray development. Instead, mesenchymal cell condensation surrounded by a thick layer composed of basement membrane and basal lamella is probably crucial for its morphogenesis at the distal tips of developing spiny rays (Fig. 4). The mesenchymal cells contain BMP signaling-positive premature osteoblasts, and an appropriate intensity of BMP signaling gives rise to the proper morphology of the spiny ray. Interestingly, in the thorny spine of *S. cirrhifer*, the lateral thorny protrusions also equip the premature osteoblast condensation, which contains BMP signaling-positive mesenchymal cells, as does the stem tip of the spiny ray. The acquisition of the thorny spine in *S. cirrhifer* may have resulted from a spatial modification of an ancestral developmental mechanism that was originally established at the tips of spiny rays. Although our observations of *S. cirrhifer* provide correlative rather than definitive evidence, the results presented here nevertheless suggest that modifications of the ancestral developmental mechanisms of spiny rays may have triggered morphological diversification of these structures in specific acanthomorph lineages (Fig. 7).

Actinotrichia are considered an essential structure for fin ray development and regeneration[28,29,32,33,37] in the growth process of soft rays, and it has been known that two hemirays are separated into right and left elements by a partition of a thick bundle of actinotrichia (Supplementary Fig. 6)[29,31,32,39,42]. In contrast, in the developmental process of spiny rays, the absence of actinotrichia may result in two hemirays fused into a stiff, single column of bone, creating a thick and strong bone. Notably, as an alternative to actinotrichia-dependent growth mechanisms, spiny-ray development probably depends on a "cap-like" structure at the tips, which contains the mesenchymal cell condensation surrounded by a thick layer composed of basement membrane and basal lamella. To elongate the spiny-ray bone, mesenchymal cells must move distally, pushing out the epithelial cell layer. This cap-like structure may benefit a process of distal movement beneath the epidermis. A previous study reported that distinct transcription factor signatures characterize the spiny- and soft-ray domains[11]. The spiny-ray domain specifically expresses *alx4a/b* and

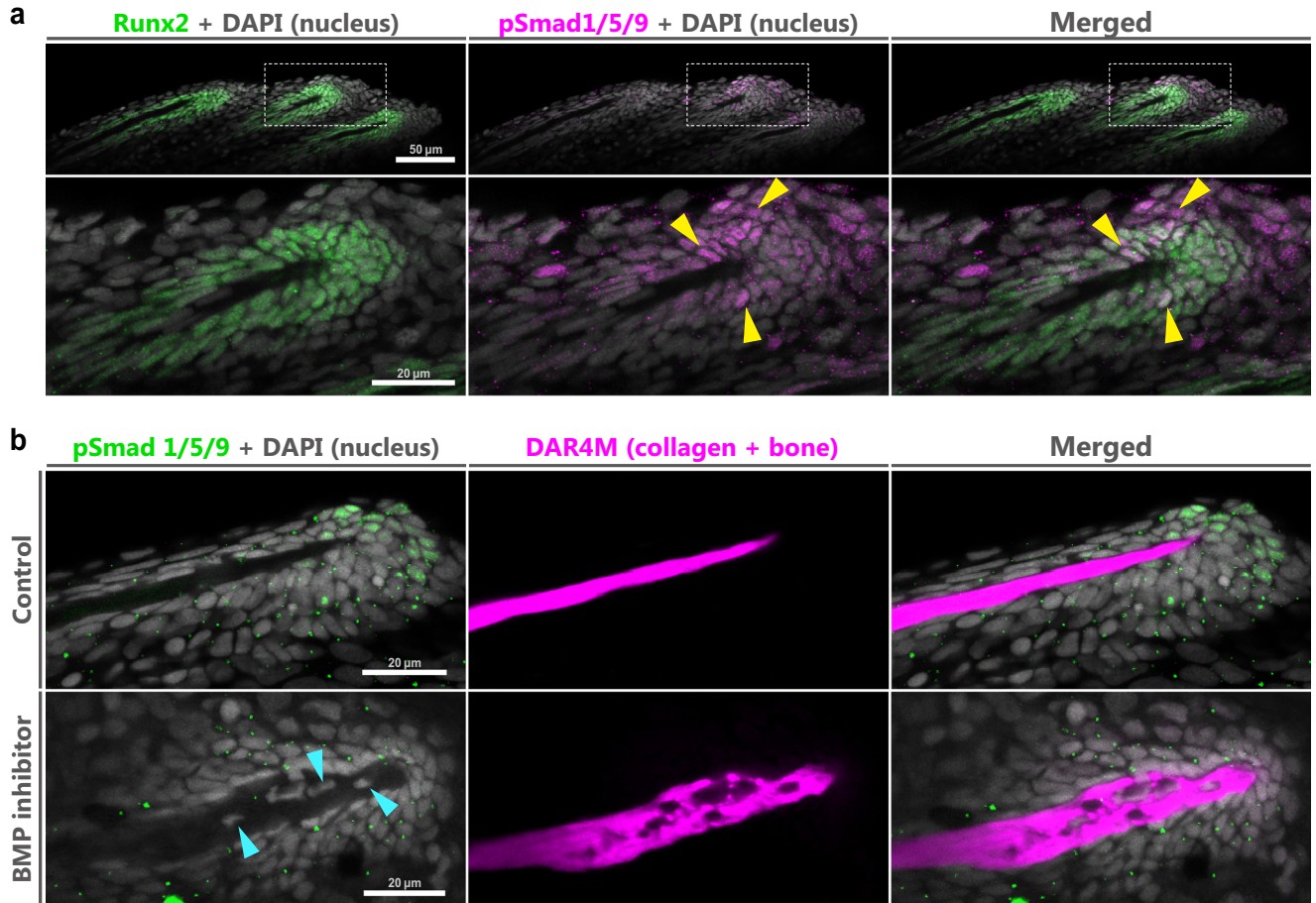

**Fig. 5 | Proper osteogenesis depends on BMP signaling at the tips of spiny rays in *Melanotaenia praecox*. a** Distributions of Runx2-positive cells (green) and pSmad1/5/9-positive cells (magenta) in the spiny rays of a DF-st4 larva stained by immunohistochemistry with DAPI (white) (*n* = 9). Yellow arrow heads show examples of pSmad1/5/9 and Runx2-positive cells. The lower panels are magnified images of the white dashed box in the upper panels. **b** pSmad1/5/9 signals (green) and bone morphologies (magenta) in a control fish (*n* = 7) and fish with inhibited BMP signaling (*n* = 8). Spiny rays in the DF-st4 larvae were stained by immunohistochemistry with DAR4M (magenta) and DAPI (white). Blue arrows indicate cells embedded in the spiny-ray bone. All images are optical sections obtained with confocal microscopy.

*tbx2b*, whereas the soft-ray domain expresses *hoxa13a/b*. Our study revealed differences in developmental mechanisms between spiny and soft rays at both cellular and ECM levels. These differences may be regulated by transcription factors that exhibit mutually exclusive expression patterns in the two domains. Consistent with this idea, ablation of *hoxa13a/hoxd13a*-expressing fin fold mesenchymal cells reduced the formation of actinotrichia. Taking these findings together with their expression specificities, *alx4a/b* and *tbx2b* may promote the mesenchymal cell aggregation characteristics of spiny rays, whereas *hoxa13a/b* may enhance the actinotrichia synthesis essential for soft rays.

Hoch et al.[11] reported that inhibition of BMP signaling prior to fin-ray formation induced homeotic transformations of prospective spiny rays into soft rays. In contrast, our DMH1 treatment, applied after the spiny rays had already formed, did not result in any apparent transformation of spiny rays into soft ray-like structures. Although we cannot entirely rule out minor shifts in osteoblast identity, the absence of segmentation or actinotrichia-like collagenous structures in treated specimens indicates that BMP inhibition did not induce a major transformation from spiny to soft rays once the spiny rays had formed. Taken together, these findings suggest that BMP signaling primarily contributes to establishing the anterior–posterior division of the dorsal and anal fins into spiny- and soft-ray domains, but may not drive a qualitative switch in osteoblast identity after cell differentiation has occurred.

Building on previous studies, our findings allow us to speculate on the evolutionary origin of spiny rays. One simple hypothesis is that spiny rays evolved from soft rays through the loss of their actinotrichia in the anterior part of each fin. However, the soft-ray morphology of the *and1*⁻/⁻; *and2*⁺/⁻ mutants did not resemble that of spiny rays because the soft rays lacked their straight morphology. Therefore, this hypothesis is not supported by the data. To definitively test the evolutionary hypothesis regarding the evolutionary acquisition of spiny rays, further studies are required on developmental mechanisms in species belonging to lineages of the teleost phylogeny that diverged earlier than the Acanthomorpha did. Furthermore, the evolution of bone types might have contributed to the acquisition of certain features of spiny rays. The majority of Euteleostei, including Acanthomorpha (except for some species such as tunas), possess acellular bone, which lacks osteocytes[59]. In contrast, more basal groups of fishes are characterized by cellular bone, which contains osteocytes[59]. Although the phylogenetic timing of the transition from cellular to acellular bone does not coincide with the emergence of spiny rays, the prior establishment of acellular bone may have facilitated the acquisition of features such as the fusion of the two hemirays and the stiff structure of spiny rays.

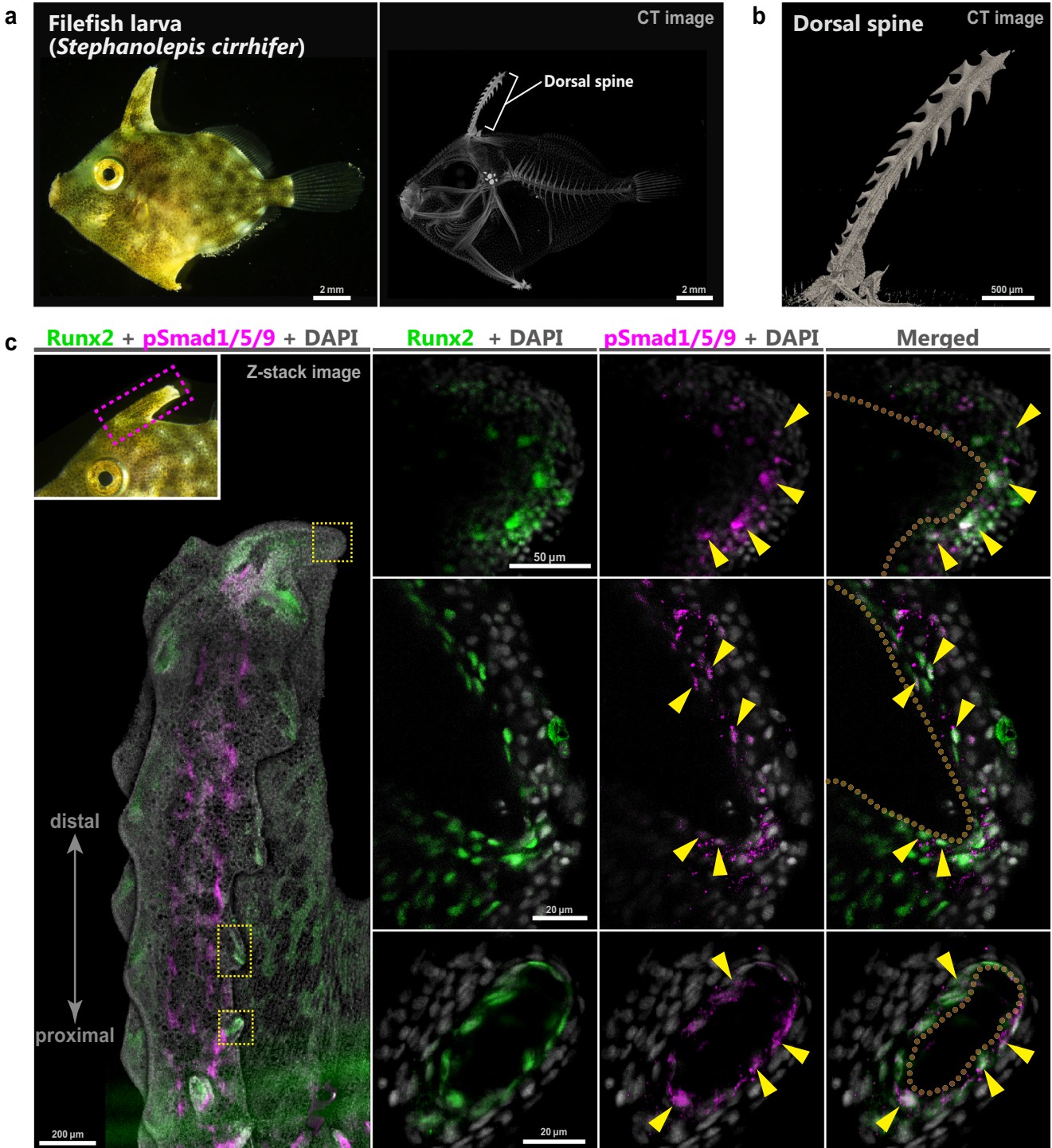

**Fig. 6 | Premature osteoblasts with BMP signaling are spatially modified from the tip to the lateral protrusions of the first dorsal-fin spiny ray in *Stephanolepis cirrhifer*. a** Lateral view of a larva, and a micro-CT scan of the skeleton (*n* = 2). **b** Magnified CT image of the dorsal spine. **c** Distributions of Runx2-positive cells (green) and pSmad1/5/9-positive cells (magenta) in the dorsal spine, stained by immunohistochemistry with DAPI (white) (*n* = 3). Upper left inset shows a magnified view of the dorsal spine region. Yellow arrow heads show examples of pSmad1/5/9 and Runx2-positive cells. Dorsal-spine bones are outlined in orange. All panels except the left-most panel represent magnified views of the area enclosed by the yellow dashed box in the left-most panel. The left-most picture is a confocal Z-stack image, and the others are optical sections obtained by confocal microscopy.

The data presented and discussed here allow us to more easily distinguish the "true spine" from other spine-like fin-ray structures that have been conventionally called either "spines" or "spiny rays"[5,11,18]—based on the developmental mechanism, such as the presence or absence of actinotrichia. Some catfish (Siluriformes) are known to have saw-like spiny rays in the pectoral fin, and the developmental process of their spiny ray resembles that of typical soft rays, which use actinotrichia[17]. It has been thought that the spiny rays of catfish were possibly acquired independently of the true spines seen in the Acanthomorpha[11]. Consistent with this idea, our findings strongly

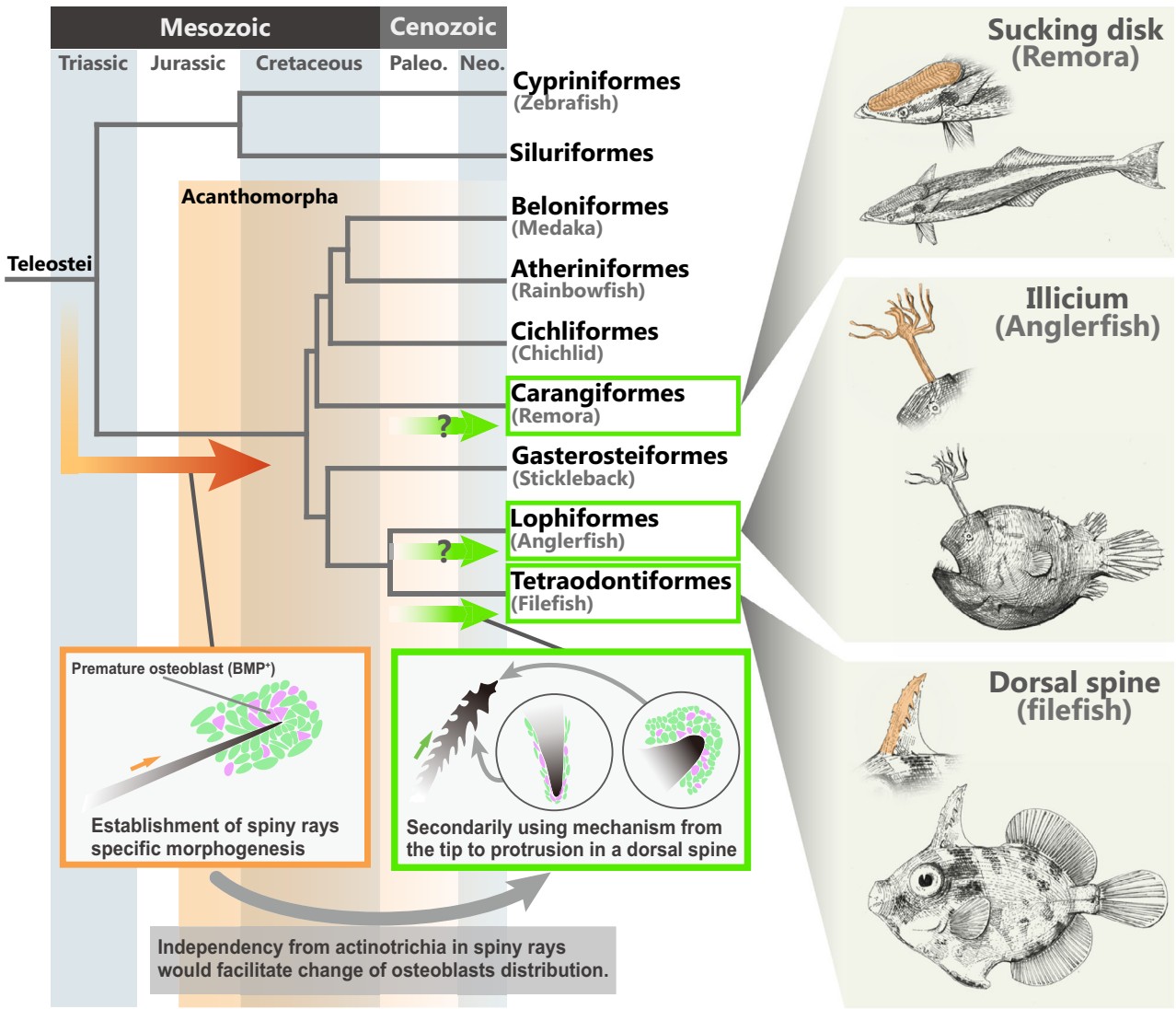

**Fig. 7 | Diversification of the spiny-ray morphology in the Acanthomorpha.** Simplified scenario of the evolution of the spiny-ray morphology. In the phylogenetic tree, the right-angled orange arrow represents the evolutionary course to spiny-ray development, and the green arrow represents the evolutionary course to modified spiny-ray development. The schematic illustrations at the bottom show a simple rod-like spiny ray in the dorsal fin of a *Melanotaenia praecox* larva (left), and the thorny spine in the filefish *Stephanolepis cirrhifer* larva (right).

suggest that the developmental mechanisms of true spines in the Acanthomorpha are completely different from those of spiny rays in catfish.

Here, we observed the distribution of Runx2-positive osteoblasts and pSmad1/5/9 in the spiny rays of *M. praecox* and the thorny spine of *S. cirrhifer*. These observations suggest that the dorsal spine of *S. cirrhifer* may have evolved by modification of the spiny-ray developmental mechanisms, which include the osteoblast distribution and BMP signaling. Our pharmacological inhibition assay in *M. praecox* demonstrated that BMP signaling in spiny rays regulates the thickness of the accumulating bony matrix (Fig. 5b). In addition, the colocalization patterns of Runx2 and pSmad1/5/9 differed between the spiny rays of *M. praecox* and the thorny spine of *S. cirrhifer*. Together, these data suggest that in the thorny spine of *S. cirrhifer*, position-dependent modifications in BMP signaling intensity may regulate thorn formation. Based on investigation of the thorny spine of *S. cirrhifer*, we could assume that similar modification may have occurred in the evolution of several Acanthomorpha fish lineages, in that some taxa possess extremely modified spiny rays, such as the sucking disc of remoras[23,24] and the fishing apparatus of goosefishes and anglerfishes[8,25] (Fig. 7).

For rod-shaped spiny rays to evolve into these highly modified structures, the distribution of osteoblasts and the intensity of bone accumulation should have been altered during their evolution. In addition, changes in BMP signaling, which plays a crucial role in spiny ray bone morphogenesis, could be considered a potential upstream factor for spiny ray modification. We anticipate that intriguing future works on the genomic changes involved in the regulation of osteoblast distribution and intensity of molecular signaling, such as BMP signaling, in these teleosts will shed light on the evolutionary mechanism of spiny-ray morphologies.

In contrast to spiny-ray diversifications, the basic morphology of the soft-ray hemirays appears to have remained unchanged. Even though the soft rays of some fish might form a hugely fan-like or long wing-like structure (e.g., the pectoral fins of lionfish and the caudal and pectoral fins of flying fish), it is simply the size and length of the fins that have changed in those modified fins. There could be several reasons why the morphologies of spiny rays are more highly diversified compared with those of soft rays. Though we could assume that some external factors, such as differences in the selection pressure, have caused the difference in the extent of diversification between spiny

and soft rays, it is very possible that internal factors, particularly developmental mechanisms, have also facilitated spiny-ray morphological diversification. When the shapes and lengths of spiny and soft rays changed, their growth direction should have changed as well. In soft rays, mesenchymal cells use the needle-shaped actinotrichia as scaffolds to generate the proper pattern of the fin bones[31,32]. In this situation, osteoblasts that depend on actinotrichia would not be reused in other positions except at the tips of soft rays, and the growth direction of soft-ray bones would be restricted to a straight line distally. Conversely, spiny rays may have evolved without this restriction because they are actinotrichia-independent. Under the developmental mechanism of spiny rays, the growth direction should be determined by the distribution of osteoblasts around the spiny-ray bones. The pre-established independence from actinotrichia and the autonomous osteogenic mechanisms at the tips of spiny rays may have been spatially modified and redeployed at alternative positions. Combined, the change in the osteoblast distribution and the function of the molecular signaling that regulates the accumulation of bony structure have probably driven the changes from a simple spine to other complex bone morphologies.

These two types of fin rays, spiny and soft rays, can be regarded as a good model to illustrate that the distribution of mesenchymal cells, the signaling-molecule intensity, and the usage of ECM such as in actinotrichia, contributed to the morphological diversification (Fig. 7). We suggest that, in addition to genome duplications[60], acquisition of regulatory cis-elements[61], and the innovation of cell types[62] and molecular interactions[63], variation in cell distribution and the usage of ECM, such as in the actinotrichia in our fish models, may also be major factors determining the extent of morphological diversification in a broad range of animal evolution.

## Methods

### Animal husbandry and embryo culture
*Melanotaenia praecox* were hatched and maintained in a laboratory at Tohoku University, as previously described[12,36]. Adults were held in 1-L or 3-L tanks, at -28 °C, a pH slightly greater than 7.0, and under a light: dark cycle of 14:10 h. Hatched larvae were transferred to a 250-mL rearing tank, at a density of 1–15 individuals per tank. Adults were fed live brine shrimp once or twice daily. Depending on the progeny size, larvae were fed live *Paramecium* at least once every 2 days, and/or brine shrimp at least once daily. Removal of dead larvae and excreta was performed as required. After the juvenile stage, the progenies were moved to larger tanks (≥ 250 mL). When observing the dorsal fin bone morphologies of young male and female fish (male, n = 3; female, n = 3), whose sexes are distinguishable by their body coloration[36], we did not find any significant differences in the fin bone morphologies between males and females at this stage (Supplementary Fig. 8). Furthermore, we did not have a reliable technique to distinguish the sex of larval and juvenile *M. praecox* using PCR-based methods or other investigations, and thus did not distinguish the sex of the larval and juvenile specimens in our experiments.

Larvae of *Stephanolepis cirrhifer* were captured at Shoubutahama Fishing Harbor in Miyagi, Japan (38°16′55″ N, 141°03′41″ E). Fish identification followed the work of Okiyama[64]. Although sex chromosome system and sexual dimorphisms in the adult second dorsal fin have been reported in this species[65,66], we did not distinguish the sex of our specimens because a reliable PCR-based genotyping method has not yet been established. According to Kwon et al.[67], sexual maturation in this species occurs at a total length of approximately 11.7 cm. The specimens used in our experiments were all significantly smaller than this size and would thus not have been expected to have developed sexual dimorphisms. Furthermore, sexual dimorphisms of the dorsal spine have not been reported. Taken together, we concluded that distinguishing the sex of our *S. chirrifer* specimens was unnecessary.

Animal care experimental procedures were conducted in accordance with institutional and national guidelines and regulations, and were approved by the Tohoku University Animal Research Committee (permit number 2022LsA-002-10). The study was carried out in compliance with ARRIVE guidelines.

### X-ray micro-CT scanning
The skeletal structures of *M. praecox* adults (Male, n = 3) and *S. cirrhifer* larvae (0.98 cm SL & 1.2 cm SL, n = 2) were observed using X-ray microcomputed tomography (CT) scanning, as previously described in Akama et al.[68]. Samples of *M. praecox* adults were fixed with 10% formaldehyde at room temperature overnight, followed by dehydration using an ethanol series (50% EtOH, 70% EtOH, 80% EtOH, 90% EtOH, 95%EtOH, 100% EtOH, 95% EtOH, 90%EtOH, 80%EtOH, 70%EtOH). These samples were then stored in 70% ethanol. Samples of *S. cirrhifer* larvae were fixed with 4% paraformaldehyde (PFA) at 4 °C overnight, and then transferred to 100% methanol. Using an X-ray micro-CT scanner (ScanXmate-D180RSS270, modified by Voxel Works Co., Tokyo, Japan and VOXIA room VR150V-HH-TAVE, Voxel Works Co., Tokyo, Japan), the fixed specimens were scanned at conditions shown in Supplementary Table 1. The micro-CT data were reconstructed using coneCTexpress (White Rabbit Corp.). Three-dimensional image analysis was performed with Molcer v.1.8.5.1 (White Rabbit Corp.).

### Small molecule staining
Live larvae were stained to examine bone[12] (double staining, DF-st4, n = 10; only calcein staining, young male, n = 3; only calcein staining, young female, n = 3) and actinotrichia[32] (WT, DF-st4 or juveniles with an SL < 1 cm, n = 8; *and1^{−/−}/and2^{+/−}*, DF-st4 or juveniles with an SL < 1 cm, n = 6; *and1^{+/−}/and2^{−/−}*, DF-st4 or juveniles with an SL < 1 cm, n = 2; *and1^{−/−}/and2^{+/+}*, DF-st4 or juveniles with an SL < 1 cm, n = 1) as described previously, with minor modifications. To observe the actinotrichia, vital staining with 5 μM of DAFFM DA (Goryo Chemical, #SK1004-01) was used. Larvae of *M. praecox* were immersed in the DAFFM DA solution overnight at room temperature. After staining of the actinotrichia and/or bone, the larvae were washed twice with system water. Thereafter, staining of bone with alizarin red was conducted. For observations, larvae were anesthetized with 0.025% MS222/E3, placed on a 1% agarose gel/E3, transferred immediately to a small case filled with system water, and awakened with water. The stained live larvae were observed and photographed under a microscope (Leica M205 FA, with Leica DFC 360 FX attachment). The images were analyzed using Leica LAS-AF software (Leica Microsystems Inc.), LAS-X software (Leica Microsystems Inc.), and Adobe Photoshop CS6.

To observe actinotrichia in fixed specimens, *M. praecox* larvae (DF-st3, n = 3; DF-st4, n = 4; juvenile, n = 4) were fixed in 4% PFA overnight at 4 °C. After washing two times in phosphate-buffered saline (PBS) with Tween 20 (PBT: 0.1% Tween 20 in PBS), specimens were stained with 10 μM of DAFFM (Goryo Chemical, #SK1003-01) overnight at room temperature, and then washed two times in PBT. To further clear and remove the pigment, the stained samples were immersed in 0.5% KOH/3% $H_2O_2$/PBT solution overnight at 4 °C, and then washed three times in PBT. The head and abdomen were dissected from the samples to prepare them for observation. The specimens were placed on a glass slide, covered with a coverslip, lightly pressed, and observed under a confocal microscope system (Leica TCS SP5 II).

### DAR4M and SYTO13 staining
To observe cells, actinotrichia, and the bone condition, *M. praecox* larvae (spiny ray, DF-st4, n = 3; soft ray, DF-st4, n = 3) were fixed in 4% PFA overnight at 4 °C, and then dehydrated, as described previously[9]. Following rehydration, specimens were immersed in 0.5% KOH/3% $H_2O_2$ solution for 30 min to 1 h at room temperature. Next, the specimens were washed three times with PBT, and stained with SYTO13

green-fluorescent nucleic acid stain (Invitrogen, #S7575) for 1 h at room temperature. After washing three times with PBT, specimens were immersed in 10 μM of DAR4M (Goryo Chemical, #SKSK1005-01) overnight at 4 °C. The next day, the specimens were again washed three times with PBT. Finally, each specimen was placed in a 1-well cell culture plate with PrimeGel Agarose LMT 1-20 K GAT (Takara Bio, 5806 A) and observed under a light-sheet microscope (QuVi SPIM, Luxendo).

## Whole-mount in situ hybridization

Whole-mount in situ hybridization for gene expression analysis was performed as previously described[69,70], with minor modifications. Samples of *M. praecox* (*and1*, DF-st4 larvae, n = 6; *and1*, juvenile, n = 4; *and2*, DF-st4 larvae, n = 4; *and2*, juvenile, n = 4) were fixed with 4% PFA/PBS at 4 °C overnight, dehydrated in methanol/PBT, and stored at −30 °C in methanol. The samples were then rehydrated progressively with 75% methanol/$H_2O$, 50% methanol/$H_2O$, 25% methanol/Tris-buffered saline with Tween 20 (TBST), and TBST, for 10 min each. Next, these samples were bleached with 3% $H_2O_2$ and 0.5% KOH in TBST for 1 h. After washing five times with TBST for 5 min, the samples were incubated in Proteinase K (Pro K, Invitrogen, #25530049) in TBST (20 μg/ml) for 1 h, rinsed once briefly with TBST, and washed once for 5 min with TBST. ProK-treated samples were placed in 0.1 M triethanolamine (TEA, pH 7–8) and incubated for 10 min after adding 10 μl of acetic acid per 1 ml TEA. After washing five times with TBST for 5 min, samples were refixed with 4% PFA/PBS solution for 20 min, and washed five times with TBST for 5 min. The samples were placed in hybridization buffer (containing 50% formamide, 5× SSC, 0.1% Tween 20, 5 mg/ml of tRNA, 50 μg/ml of heparin, 2.5% blocking regent, and 10 mM of EDTA) at 65 °C for 5 h, and then incubated in hybridization buffer containing 5–10 μg/ml of probe, overnight, at 65 °C. Probe-hybridized samples were washed twice with 50% formamide/5× SSC/0.1% Tween 20/10 mM of EDTA at 65 °C for 30 min, three times with 2× SSC/0.1% Tween 20 at 65 °C for 20 min each, and three times with 0.2× SSC/0.1% Tween 20 at 65 °C for 20 min each. After washing twice with maleic acid buffer for 5 min, and blocking with blocking buffer (1% blocking regent/10% heat-inactivated goat serum in maleic acid buffer) for 1.5 h, the samples were incubated in blocking buffer containing 1:3000 anti-digoxigenin-AP Fab fragments (Roche #11093274910) for 5 h. These samples were washed with maleic acid buffer as follows: twice for 5 min, three times for 20 min, once overnight, four times for 2 h, and once overnight. After washing three times with TBST for 5 min, the samples were incubated in AP buffer (0.1 M Tris base at pH 9.5, 0.1 M NaCl, 0.1% Tween 20). The samples were stained with BM-Purple (Roche #11442074001) in the dark, at room temperature, for 1–7 days. The staining reaction was stopped by washing the samples three times for 10 min with PBT. All steps were performed on a shaking platform.

For in vitro transcription of the digoxigenin (DIG)-labeled RNA probe, the polymerase SP6 (Roche #10810274001) or T7 RNA (Invitrogen #18033019) was used, along with a DIG RNA labeling mix (Roche #11277073910), followed by a deoxyribonuclease step and ethanol precipitation. The labeled in situ RNA probes were resuspended in 50% formamide and stored at −30 °C.

## Molecular cloning and sequencing

Molecular cloning and sequencing were performed following the protocol outlined in a previous study[36], with slight modifications. A primer table is provided in Supplementary Table 2. *Melanotaenia praecox* sequences were identified by BLAST (Basic Local Alignment Search Tool) against genome data for Boeseman's rainbowfish, *Melanotaenia boesemani*.

## Generation of CRISPR/Cas9 and1/and2 mutant M. praecox lines

To investigate the functions of the actinotrichia in vivo, CRISPR/Cas-induced knockout fish were generated. In accordance with a previously described method[36], four types of sgRNAs (Supplementary Figs. 2 and 3, and Supplementary Table 3) for CRISPR/Cas were prepared, and microinjection into *M. praecox* fertilized eggs was performed (Supplementary Fig. 9a). We prepared a 5-μl solution containing 100 ng of each sgRNA, 1,250 ng of Cas9 protein, and 0.5 μl of phenol red in nuclease-free water, and then 2–3 nL of this solution were injected into the single cell of the embryo.

The injected fish ($G_0$) with fin bone phenotype were in-crossed to produce a generation of homozygous *and1* knockout (*and1$^{-/-}$*) and heterozygous *and2* knockout (*and2$^{+/-}$*) fish ($F_1$) (Supplementary Fig. 9). Although the homozygous *and1/and2* knockout fish (*and1$^{-/-}$; and2$^{-/-}$*) was the most ideal for this study, this genotype could not be reared, probably because this knockout mutation is lethal or causes a high mortality rate. To determine the sequences of mutant alleles, the genomic region, including the target site of sgRNAs was amplified with KOD One® PCR Master Mix -Blue- polymerase (Toyobo #KMM-201) using the primer pairs listed in Supplementary Table 2. The PCR conditions were as follows: one cycle at 94 °C for 2 min, followed by 35 cycles of 98 °C for 10 sec, 60 °C for 5 sec, and 68 °C for 15 sec. Obtained fragments were cloned using a TOPO Cloning Kit (Invitrogen #K460001) or TArget Clone™ -Plus- (TAK-201) and sequenced by Sanger sequencing. Following the genotyping conducted by the above sequence analysis, after observation and genotyping of the $F_1$ fish, we in-crossed selected single pairs of $F_1$ fish that had fin phenotype and *and1$^{-/-}$/and2$^{+/-}$* genotypes, and we obtained *and1$^{-/-}$/and2$^{+/-}$* fish ($F_2$) (Supplementary Fig. 9a). Some of the $F_2$ fish obtained from single pairs of $F_1$ fish had a few short actinotrichia in their fins (Supplementary Fig. 9b) and we confirmed that one of these fish genotypes was homozygous *and1* knockout only (*and1$^{-/-}$/and2$^{+/+}$*) (Supplementary Fig. 9c). This analysis suggests that homozygous *and1* knockout alone does not cause the actinotrichia loss. Then, we only used fish that lost radially arranged actinotrichia as in wild type fish and had *and1$^{-/-}$/and2$^{+/-}$* genotypes for our analysis (Fig. 2b, c). The sequences of mutant alleles of $F_2$ fish using observation were also determined by the above sequence analysis. Detailed information on the *and1/and2* mutations in the $F_1$ and $F_2$ fish is summarized in Supplementary Figs. 2 and 3. To further analyze the *and1$^{+/-}$; and2$^{-/-}$* fish, we first outcrossed $F_1$ individuals with the genotype *and1$^{-/-}$; and2$^{+/-}$* to wild-type fish, obtaining $F_2$ fish with the genotype *and1$^{+/-}$; and2$^{+/-}$* (Supplementary Fig. 10). Next, we crossed two types of $F_2$ individuals—those with *and1$^{-/-}$; and2$^{+/-}$* genotypes, and those with *and1$^{+/-}$; and2$^{+/-}$* genotypes—to generate F3 offspring. For our analyses, we used only F3 fish that had the *and1$^{+/-}$; and2$^{-/-}$* genotype (Supplementary Fig. 10). The mutant alleles of the analyzed F3 fish were confirmed by sequence analysis, and detailed information on the *and1* and *and2* mutations is summarized in Supplementary Fig. 5b, c. In this study, because some alleles with missense mutations are not expected to encode a nonfunctional truncated protein, we denote those alleles with the symbol "+".

## TEM analysis

Preparation of epoxy blocks was performed as described previously[71], with minor modifications. Larvae of *M. praecox* (spiny ray, DF-st4, n = 3; soft ray, DF-st4, n = 3) were fixed in 2% glutaraldehyde and 2% paraformaldehyde in 0.01 M PBS at 4 °C overnight. The fixed specimens were post-fixed in 1% OsO4 in 0.01 M PBS at 4 °C overnight, after decalcification in 10% ethylenediaminetetraacetic acid in 0.01 M PBS at 4 °C for 4 days. The specimens were then embedded in epoxy resin (Epon 812; TAAB Laboratories Equipment Ltd) by polymerizing overnight at 37 °C, and again overnight at 60 °C after dehydration through a graded ethanol series followed by a transition in propylene oxide.

The epoxy blocks were trimmed, and serial sections (70 nm in thickness) were cut with an ultramicrotome (Leica, EM UC7) using a Histo Diamond Knife (Syntek, SYM3045 Ultra). The sections were picked up and mounted on a grid (Veco, Square Mesh Handle 150 mesh

Nickel Grid). Next, the sections were stained with 4% uranyl acetate for 15 min and then washed three times with Mili-Q water. After drying, they were stained with lead citrate staining solution (2% w/v trisodium citrate dihydrate, 1% w/v lead (II) citrate trihydrate, 1% w/v lead (II) acetate, 1% w/v lead (II) nitrate in 0.18 mol/L sodium hydroxide solution) and then washed three times with Mili-Q water. Finally, the specimens were observed under a transmission electron microscope (JEOL, JEM1400).

## Immunohistochemistry

A whole-mount immunohistochemistry protocol was used to detect laminin (Sigma-Aldrich #L9393), TP63 (Abcam, #ab735), Runx2 (Santa Cruz Biotechnology, #sc-101145), and pSmad1/5/9 (Cell Signaling Technology, #13820). Whole-mount immunohistochemistry was performed as described previously[9], with minor modifications, until primary antibody incubation steps were different between staining laminin and TP63, and staining Runx2 and/or pSmad1/5/9.

To stain laminin and TP63, *M. praecox* larvae (DF-st4, $n = 9$) were fixed with absolute methanol. The samples were rehydrated with methanol/PBT, blocked with blocking buffer (1% BSA, 1% goat serum, and 1% DMSO in PBS) and stained with a 1:50 dilution of anti-laminin antibody and 1:50 dilution of anti-TP63 antibody in blocking buffer overnight. Samples were then washed five times in PBT, blocked with blocking buffer, and incubated with a 1:500 dilution of Alexa Fluor 488 goat anti-rabbit (Invitrogen, #A-11008) and Alexa Fluor 594 goat anti-mouse (Invitrogen, #A-11032). After washing five times in PBT, samples were stained with a 0.001% w/v DAPI (4′,6-Diamidino-2-phenylindole dihydrochloride; Dojindo, #342-07431) for 1 h, and then washed three times in PBT. The head and abdomen were dissected from the samples to prepare them for observations. The samples were placed on a glass slide, covered with a coverslip, lightly pressed, and observed under a confocal microscope system (Leica TCS SP5 II).

To stain Runx2 and/or pSmad1/5/9, *M. praecox* (DF-st4, $n = 9$) and *S. cirrhifer* larvae (1.2–1.35 cm SL, $n = 3$) were fixed with 4% PFA/PBS, dehydrated with methanol/PBT, and stocked in absolute methanol at −20 °C. Following incubation in absolute acetone for 20 min, the samples were rehydrated with methanol/PBT, permeabilized with 1% TritonX-100, blocked with blocking buffer (1% BSA, 1% goat serum, and 1% DMSO in PBT) and stained with a 1:50 dilution of anti-Runx2 antibody and/or 1:100 dilution of anti-pSmad1/5/9 antibody in blocking buffer overnight. Secondary antibody and DAPI staining procedures are shown above, and Alexa Fluor 488 goat anti-rabbit (Invitrogen, #A-11008), Alexa Fluor 488 goat anti-mouse (Invitrogen, #A-11001), and Alexa Fluor 594 goat anti-rabbit (Invitrogen, #A-11012) were used as the secondary antibodies. DMH1-treated specimens (DMH1 treated, DF-st4, $n = 7$; control, DF-st4, $n = 8$) were additionally stained with DAR4M after the DAPI staining.

## DMH1 treatment experiments

Inhibition of BMP signaling was performed with 5 µM of DMH1 (Selleckchem #S7146, dissolved in DMSO to 20 mM). DMH1 stock solution was dissolved in E3 water[72]. The *M. praecox* larvae (DF-st4, $n = 7$) were treated with DMH1 in the dark at 28 °C for 10 days, subsequently washed with system water three times, and fixed with 4% PFA for phenotype analysis. Mock treatments (DF-st4, $n = 8$) were performed using 0.025% DMSO, which does not result in phenotypic changes.

## Reporting summary

Further information on research design is available in the Nature Portfolio Reporting Summary linked to this article.

## Data availability

The CT data generated in this study have been deposited in the figshare database under the accession code: https://doi.org/10.6084/m9.figshare.30744728. All unique biological materials generated in this study, including mutant lines of *Melanotaenia praecox*, are available from the corresponding author upon reasonable request. Some wild-caught specimens of *Stephanolepis cirrhifer* are subject to local collection and transfer regulations and therefore cannot be freely distributed.

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

## Acknowledgements

We are grateful to members of the Tamura Laboratory (Laboratory of Organ Morphogenesis) for assistance with fish maintenance and experimental techniques. We thank Kaori Karashimada, a technical staff member at the Center of Common Research Laboratory, Graduate School of Dentistry, Tohoku University, for performing TEM image acquisitions. We also thank Dr. Harumasa Kano (technical staff member at the Tohoku University Museum) and White Rabbit Corporation for performing CT image acquisitions. This study was supported by JSPS KAKENHI (grant nos. JP21K19202 21H05768, JP22H02627, 23H04301, 23KJ0206 and 25H01423).

## Author contributions

K.M. and K.T. conceived the project and wrote the manuscript. K.M. conducted the experiments, collected live *Stephanolepis cirrhifer*, and analyzed the data. J.K. established the method to visualize collagen. S.A. helped to establish the procedures for genome editing in *Melanotaenia praecox*. S.K. and N.F. provided the light-sheet microscopy imaging infrastructure. Y.S. provided the transmission electron microscopy analysis infrastructure. J.K., S.A., G.A., and M.U. provided critical comments for improving the manuscript.

## Competing interests

The authors declare no competing interests.

## Additional information

[1]Department of Ecological Developmental Adaptability Life Sciences, Graduate School of Life Sciences, Tohoku University, Sendai, Miyagi, Japan. [2]Laboratory of Organ Morphogenesis, JT Biohistory Research Hall, Osaka, Japan. [3]Graduate School of Science, Osaka University, Osaka, Japan. [4]Department of Environment and Sustainability, School of Environment and Sustainability, Mukogawa Women's University, Nishinomiya, Hyogo, Japan. [5]Division of Craniofacial Development and Tissue Biology, Graduate School of Dentistry, Tohoku University, Sendai, Miyagi, Japan. [6]Division of Developmental Biology, Department of Functional Morphology, School of Life Science, Faculty of Medicine, Tottori University, Tottori, Japan. [7]Ushimado Marine Institute (UMI), Okayama University, Setouchi, Okayama, Japan. [8]Department of Biophysics, Graduate School of Science, Kyoto University, Kitashirakawa-Oiwake, Sakyo-ku, Kyoto, Japan. ✉e-mail: kazuhide.miyamoto.t5@dc.tohoku.ac.jp

