## [Peer Review file · Nature Communications]

Actinotrichia-independent developmental mechanisms of spiny rays facilitate the morphological diversification of Acanthomorpha fish fins

Corresponding Author: Mr Kazuhide Miyamoto

Version 0:

Reviewer comments:

Reviewer #1

(Remarks to the Author)

In this study, Miyamoto et al. demonstrated that the developmental mechanisms of spiny rays in Acanthomorpha fish are independent of actinotrichia, in contrast to those of soft rays, by introducing the rainbowfish *Melanotaenia praecox* as a new model organism. Through transmission electron microscopy (TEM) analysis and immunostaining, the authors identified the condensation of mesenchymal cells and the presence of premature Runx2/pSmad-positive osteoblasts at the tips of spiny rays. Additionally, they found that Runx2/pSmad-positive cells were similarly distributed in the tips and lateral protrusions of spiny rays in *Stephanolepis cirrhifer*. Regarding the authors' primary claim presented in the title—that actinotrichia-independent developmental mechanisms “facilitate” morphological diversification in Acanthomorpha fish—I feel that the supporting results appear to provide correlative evidence rather than strong proof. However, this study offers valuable insights into the developmental mechanisms of spiny rays, which contribute to the highly diversified morphology observed in some species; nevertheless, the following points should be addressed further.

In situ hybridization reveals slight expression of *actinodin2* but no detectable expression of *actinodin1* in the developing spiny rays of *Melanotaenia praecox* (Extended Fig 1). The *and1*^{-/-} and *and2*^{+/-} phenotype is presented in this manuscript; however, there is no description of homozygous mutants of *and2*. Including the *and2*^{-/-} or *and1*^{+/-} and *and2*^{-/-} phenotypes would further clarify the independence of actinotrichia, which is one of the main findings of the paper.

The phenotype of the *and1*^{-/-} and *and2*^{+/-} mutants is intriguing. The authors briefly noted that the soft rays are deformed in the mutants; however, could a more detailed analysis be conducted? In the mutants, the soft rays do not appear to be segmented. Additionally, is the elongation of the soft rays affected in the mutants? Why do they curl up in the caudal fin? I understand this is somewhat tangential to the main topic of the paper, but a more detailed analysis could yield new insights into soft ray formation by actinotrichia.

Regarding Figure 5b, a previous study by Hoch et al. (PNAS 2021) demonstrated that DMH1 treatment causes spiny rays in cichlid fish to transform into soft rays. Perhaps the DMH1 treatments were carried out after the initial development of the spiny ray, but that should be properly described. In this study, larvae were incubated for 10 days, which is a considerable duration. Can the authors rule out the possibility that DMH1 treatments caused the spiny ray to develop a soft ray-like structure, resulting in the observed deformation? Do DMH1-treated spiny rays retain the properties of spiny rays?

Additionally, I believe the study by Hoch et al. is highly relevant to this paper and influences the authors' work. While their paper is cited in this manuscript (11), a more detailed description is needed. Their findings, which illustrate the establishment of spiny and soft-rayed fin domains in acanthomorph fishes, provide essential molecular insights into the differences between spiny and soft-rayed fins.

In Figure 6, the Runx2/pSmad signals largely overlap in the regions surrounding the tips of the laterally protruding spiny ray in filefish. In contrast, Figures 5a and 5b, which utilize *Melanotaenia praecox*, show less overlap in the Runx2/pSmad signals, particularly at the tips of the spiny fins. Could this discrepancy be attributed to the presentation of optical sections? The images should be updated to display the z-stack images to clearly show the distributions of the pSmad signals.

How many larvae of *Stephanolepis cirrhifer* were examined using immunostaining? The staining results from other individuals should be presented as supplementary data to confirm reproducibility.

Regarding the evolution of spiny rays, do the authors think they evolved from soft rays by losing actinotrichia, or were they acquired independently?

(Minor points)

In Fig. 1a, spiny and soft rays stained with Alizarin red are shown. Please specify which fin rays belong to which fish.

Lines 530-532: Was 5 μ l injected into each embryo? That seems like a large volume; please confirm.

Figure 5b: Are the green signals on several dots non-specific? Please clarify this point.

Line 210: While both DAFFM and DAR4M are claimed to detect actinotrichia, I am unclear about the differences between them.

Reviewer #2

(Remarks to the Author)

The manuscript entitled "Actinotrichia-independent developmental mechanisms of spiny rays facilitate the morphological diversification of Acanthomorpha fish fin", submitted to Nature Communications, investigates developmental differences between spiny rays and soft rays in rainbowfish (*Melanotaenia praecox*), with additional comparative studies from the highly modified spiny rays of the filefish (*Stephanolepis cirrhifer*).

To address their objectives, the authors employ a comprehensive suite of techniques, including micro-CT, electron microscopy, immunofluorescence, and in-situ hybridization, to demonstrate key distinctions in fin ray development and anatomy at different scales (from the entire spine to the cellular level). Specifically, they show that spiny rays are formed by an accumulation of mesenchymal cells where BMP signaling plays an important role, which differs from soft rays that rely on actinotrichia as a structural scaffold. Furthermore, the authors identify similar spiny-ray developmental mechanisms in the filefish, suggesting that this actinotrichia-independent mode of growth may have facilitated the morphological diversification of fin rays within Acanthomorpha.

Overall, the manuscript is well-executed and clearly written, with well-organized flow of ideas. The experimental design is robust to address the question proposed (although see general comments about sample size and sexes used), and the integration of the multiple methods provides a holistic and convincing analysis. The figures clearly show the results and are easy to follow. The research question is original, particularly because little is known about the skeletal development of acanthomorphs. Therefore, this research is of interests across multiple fields, including fish biology, physiology, skeletal development, and evolution. However, I suggest the authors to revise my general comments regarding this topic.

My main concern is the lack of discussion and consideration of the bone type the authors are studying (acellular bone) as well as comparing to (cellular bone), which I consider that is crucial to understand the results and do an accurate comparative analysis. I hope my review will help further improve the clarity and impact of this work.

General comments

Acellular bone skeletogenesis.

One aspect that stood out to me is that the authors do not mention, or consider, the type of bone that forms the skeleton of both species of fishes that are studied here. The majority of Neoteleostei, and thus acanthomorphs (except some species like tunas) have acellular bone (i.e. a bone without osteocytes). This type of bone is also found in some Euteleostei. In contrast, more basal groups of fishes are characterized by cellular bone (i.e. a bone with osteocytes), including Ostariophysi where cypriniformes (zebrafishes) and siluriformes (catfishes) are part of.

This difference in bone type could be one of the reasons why the development of spiny rays in zebrafish and catfish (species that authors use to compare their results) may differ from that in the species studied here. Therefore, the spiny ray development described in the manuscript may probably be related with the periosteal osteogenesis of acellular bone. Check also the anatomical terminology used (e.g. basement membranes), as some of the structures may have been already described in Moss (1961). This point can be a very interesting topic to include and discuss, since the formation of acellular bone has not been largely studied.

For these reasons, I suggest that the authors consider and discuss the acellular bone formation in the manuscript, as well as to be cautious in describing this developmental process as "unique" to Acanthomorph spiny rays, as what the authors may be describing reflect general features of acellular bone skeletogenesis rather than a distinct or novel pathway of spiny rays. Additionally, the importance of BMP signaling, while certainly relevant, is not surprising as this superfamily of proteins is well known to have a crucial role during bone development across vertebrates.

Below I included several papers that discuss the differences between acellular or cellular bone that the authors may find useful.

Sample size.

The authors specify sizes in each section of the Results, but not in Methods. I recommend also including a detailed breakdown in the Methods section, indicating the number of animals used at each developmental stage and for each technique (e.g. How many *S. cirrhifer* and *M. praecox* where μ CT-scanned?, this is not specifically clarified anywhere). Additionally, in some instances, it is unclear which ontogenetic stage is used for a given specific technique (larvae,

juveniles, adults?). Lastly, in some cases it is unknown whether the reported sample size in Results refers to the number of rays examined or the number of individual animals. Since multiple rays can be studied from a single specimen, it would be helpful to clarify the number of animals studied similar to how it is done in L130.

Animal sex.

The authors do not specify whether the specimens used were males or females. Several teleosts, including *M. preacox*, are sexual dimorphic, with males exhibiting enlarged caudal spines.

Specific comments:

Abstract

- L22. The hyphenation of “spiny rays” and “soft rays” varies throughout the manuscript. In some instances, the terms are hyphenated (e.g., lines 23, 24, 27), while in others they are not (e.g., lines 22, 48, 49, 56). Please check for consistency and revise accordingly.

- L22. “their bone structure” is a vague descriptor in this context. What about “their supporting skeletal structure”.

- L30. “the lateral protrusions equip the BMP positive osteoblast condensation...”. The verb “equip” here is a bit confusing. What about: “the lateral protrusions are associated with BMP-positive osteoblast condensation”

- L36. “... ECM usage would be major factors driving the morphological diversification in animals”. I recommend focusing this sentence specifically on acanthomorphs, rather than using the broader term “animals”.

Introduction

- L48. “...and mostly branching...”. Maybe replace “mostly branching” to “typically branched”.

- L54 – 55. Are the sucking disc of remoras or the illicium considered “spiny rays”? or they have been so modified that they are no longer considered as such?

- L72. “...there anal fin.” There is a typo here, it should be “their anal fin”.

- L78. “...with that of soft-ray development.” Those of soft-ray development instead, to be consistent with the plural subject (“matrix and cell dynamics”).

Results

- L92 – 94. This sentence reads awkwardly and may be either missing some information or be poorly structured. I recommend revising it for clarity, as its current form makes the intended meaning difficult to follow.

- L114, Figure 1. It is “yellow” instead of “white”.

- L128. “DAFFM is a small fluorescent molecule that broadly used for...”. Instead: “DAFFM is a small fluorescent molecule that has been broadly used for...”.

- L131. Delete “obviously”.

- L132. Regarding the statement “fine actinotrichia were present at the future position of spiny rays in the median fin fold”. This statement is not clearly supported by the figure. In the image (Fig 2a), the actinotrichia do not appear to be specifically localized at the future position of the spiny rays. In fact, it doesn't look like they are found in a specific position. I recommend clarifying the figure or revising the interpretation in the text to better reflect what is shown.

- L135 – 136. In the spiny and soft rays, the DAFFM only stains actinotrichia? If not, which molecules form the brightest bulk of the rays? See image below:

- L209: “n=3”. The number of spiny tails (sample size) sampled are from 3 different animals?

- L213: “...such condensed mesenchyme cells”. How the authors know that these are mesenchyme cells if they did not use any markers, only the staining of the nuclei? If the authors are defining mesenchyme cells based on their rounder, larger morphology, they could be also osteoblasts or other precursor cell?

- Figure 3: There is a typo in SYTO13 (nucleus).

Discussion

- L334: using the “unique” adjective here is unnecessary. The authors do not know if this mechanism is unique to spiny rays or to acanthomorphs, since this process can be happening in other skeletal elements or in other clades (perhaps is how the acellular bone is formed?).

- L344: Secondly?

- L361 – L363: The fact that the developmental process that forms the spiny rays described in this manuscript is different of the process of the formation of spiny rays in Siluriformes may be because catfishes have cellular bone (see my comments above).

- L406 – L411: this text looks like doesn't belong here?

Methods

- Add sample sizes for each technique (see my comment above).

- L445: after fixation, where the samples directly transferred to 70% ETOH? Or the authors dehydrated them using increasing concentration steps?

References

- Davesne, D., Meunier, F. J., Schmitt, A. D., Friedman, M., Otero, O., & Benson, R. B. (2019). The phylogenetic origin and evolution of acellular bone in teleost fishes: insights into osteocyte function in bone metabolism. *Biological Reviews*, 94(4), 1338-1363.

- Shahar, R., & Dean, M. N. (2013). The enigmas of bone without osteocytes. *BoneKEy reports*, 2, 343.

- Moss, M. L. (1961). Osteogenesis of acellular teleost fish bone. *American Journal of Anatomy*, 108(1), 99-109.

- Weiss, R. E., & Watabe, N. (1979). Studies on the biology of fish bone. III. Ultrastructure of osteogenesis and resorption in osteocytic (cellular) and anosteocytic (acellular) bones. *Calcified tissue international*, 28(1), 43-56.

- Moss, M. L. (1963). The biology of acellular teleost bone. *Annals of the New York Academy of Sciences*, 109(1), 337-350.

- Dean, M. N., & Shahar, R. (2012). The structure-mechanics relationship and the response to load of the acellular bone of neoteleost fish: a review. *Journal of Applied Ichthyology*, 28(3), 320-329.

Version 1:

Reviewer comments:

Reviewer #1

(Remarks to the Author)

The authors have responded to my comments thoroughly and satisfactorily. This is an excellent study that deserves publication in Nature Communications, and I commend the authors for their sustained efforts over the past several months.

Reviewer #2

(Remarks to the Author)

This is the second time that I have reviewed the manuscript entitled "Actinotrichia-independent developmental mechanisms of spiny rays facilitate the morphological diversification of Acanthomorpha fish fins", submitted to Nature Communications. The authors have adequately addressed my previous comments. In particular, they have responded to my suggestions regarding the discussion of cellular versus acellular bone, and they have clarified the sex of the specimens used whenever possible.

Overall, I think the authors present a robust and carefully conducted study. This work offers valuable insights into the developmental mechanisms of spiny and soft rays which, in case of spiny rays, likely contribute to the highly diversified morphologies observed in some acanthomorph species.

I have noted a few minor grammatical errors throughout the manuscript, but aside from these minor issues, the manuscript reads well and is scientifically sound.

- L50: "Branched" instead of "branching"
- L248: "We found that mesenchymal cells condensed at the tips of spiny rays, whereas such condensed mesenchymal cells were not detected at the tips of soft rays (Fig. 3a)". Personally, I don't see the condensation of mesenchymal cells in Figure 3a.
- L254: "Zebrafish" instead of "Zebrafiah"
- L280: "a Cell, ECM, and bone condition". The ECM is not particularly visible in a.
- Small yellow and pink numbers, present in figure 3 and Extended Data figure 6, that indicate regions located at the top-left corner. These are very difficult to see, maybe adding a black background may help.
- L360: "Larva" is singular, therefore it should be "has".
- L411: "Alternative to" instead of "Alternative of"

We thank you very much for reviewing our manuscript. We have revised it following careful examination of your suggestions. Your comments helped us to greatly improve the quality of our paper, and we express our gratitude to both of you for your constructive suggestions.

To improve the overall readability, we have had the entire document professionally proofread.

In addition to responding to your comments, we found some mistakes in the explanation of the phylogenetic tree in the original manuscript. Therefore, we have revised the legend of Figure 1 and Figure 2 as below:

Line 112-113, “b Simplified phylogenetic tree of the Teleostei (modified from Figs. 1 and 2 in Ghezelayagh et al.¹⁹ and Fig. 2 in Hughes et al.³⁹)”

Line 191-192, “All panels except the top-left panel represent magnified views of the area enclosed by the yellow dashed box in the top-left panel.”

In addition, we have also revised the product number of the staining molecule in our Methods section, as below:

Line 572-573, “To observe the actinotrichia, vital staining with 5 μ M of DAFFM DA (Goryo Chemical, #SK1004-01) was used.”

Please find our responses to each comment from the reviewers below.

Reviewer #1 (Remarks to the Author):

Reviewer comment 1-1:

In this study, Miyamoto et al. demonstrated that the developmental mechanisms of spiny rays in Acanthomorpha fish are independent of actinotrichia, in contrast to those of soft rays, by introducing the rainbowfish Melanotaenia praecox as a new model organism. Through transmission electron microscopy (TEM) analysis and immunostaining, the authors identified the condensation of mesenchymal cells and the presence of premature Runx2/pSmad-positive osteoblasts at the tips of spiny rays. Additionally, they found that Runx2/pSmad-positive cells were similarly distributed in the tips and lateral protrusions of spiny rays in Stephanolepis cirrhifer. Regarding the authors' primary claim presented in the title—that actinotrichia-independent developmental mechanisms “facilitate” morphological diversification in Acanthomorpha fish—I feel that the supporting results appear to provide correlative evidence rather than strong proof. However, this study offers valuable insights into the developmental mechanisms of spiny rays, which contribute to the highly diversified morphology observed in some species; nevertheless, the following points should be addressed further.

Response 1-1:

We thank the referee for the positive comments. We agree that our results provide correlative evidence rather than strong proof. Therefore, we have added a sentence to address this point.

Line 402-405, “Although our observations of *S. cirrhifer* provide correlative rather than definitive evidence, the results presented here nevertheless suggest that modifications of the ancestral developmental mechanisms of spiny rays may have triggered morphological diversification of these structures in specific acanthomorph lineages (Fig. 7).”

Reviewer comment 1-2:

In situ hybridization reveals slight expression of *actinodin2* but no detectable expression of *actinodin1* in the developing spiny rays of *Melanotaenia praecox* (Extended Fig 1). The *and1*^{-/-} and *and2*^{+/-} phenotype is presented in this manuscript; however, there is no description of homozygous mutants of *and2*. Including the *and2*^{-/-} or *and1*^{+/-} *and2*^{-/-} phenotypes would further clarify the independence of actinotrichia, which is one of the main findings of the paper.

Response 1-2:

We thank you for these helpful suggestions. To address these concerns, we generated *and1*^{+/-}; *and2*^{-/-} mutant fish and examined their fin bone morphology. We did not observe any differences in the spiny rays between *and1*^{+/-}; *and2*^{-/-} mutants and wild-type fish. Our *and1*^{+/-}; *and2*^{-/-} mutant data suggest that *actinodin2* is unlikely to play a crucial role in spiny-ray morphogenesis. We have revised the manuscript to include this description and discussion.

Line 169-172, “Furthermore, we generated *M. praecox* mutants with the genotype *and1*^{+/-}; *and2*^{-/-} (DF-st4 or juveniles with SL < 1 cm, n = 2) (Extended Data Fig. 5 and Fig. 10). In these mutants, we observed the actinotrichia were radially arranged and that the soft rays were slightly distorted (white arrowhead in Extended Data Fig. 5).”

Extended Data Figure 5 Actinotrichia distribution and spiny- and soft-ray bone morphology in the *actinodin1*^{+/-}/*actinodin2*^{-/-} knockout *Melanotaenia praecox*.

White arrowheads indicate abnormal bending of the soft rays in the knockout fish. Actinotrichia were labeled with DAFFM DA (green), and the spiny- and soft-ray bones were labeled with

alizarin red (magenta). All panels except the top-left panel represent magnified views of the area enclosed by the yellow dashed box in the top-left panel.

Line 671-678, “To further analyze the *and1*^{-/-}; *and2*^{-/-} fish, we first outcrossed F₁ individuals with the genotype *and1*^{-/-}; *and2*^{+/-} to wild-type fish, obtaining F₂ fish with the genotype *and1*^{-/-}; *and2*^{+/-} (Extended Data Fig. 10). Next, we crossed two types of F₂ individuals—those with *and1*^{-/-}; *and2*^{+/-} genotypes, and those with *and1*^{+/-}; *and2*^{+/-} genotypes—to generate F₃ offspring. For our analyses, we used only F₃ fish that had the *and1*^{+/-}; *and2*^{-/-} genotype (Extended Data Fig. 5). The mutant alleles of the analyzed F₃ fish were confirmed by sequence analysis, and detailed information on the *and1* and *and2* mutations is summarized in Extended Data Figs. 5b and 5c.”

Extended Data Figure 10. Schematic illustration of the obtaining the *actinodin1*^{+/-}/*actinodin2*^{-/-} knockout fish.

Reviewer comment 1-3:

The phenotype of the *and1*^{-/-} and *and2*^{+/-} mutants is intriguing. The authors briefly noted that the soft rays are deformed in the mutants; however, could a more detailed analysis be conducted? In the mutants, the soft rays do not appear to be segmented. Additionally, is the elongation of the soft rays affected in the mutants? Why do they curl up in the caudal fin? I understand this is somewhat tangential to the main topic of the paper, but a more detailed analysis could yield new insights into soft ray formation by actinotrichia.

Response 1-3:

We apologize for the insufficient explanation of the phenotypes of the *and1*^{-/-} and *and2*^{+/-} mutants. We are currently examining the function of the *actinodin* genes in fin-ray morphogenesis, and we plan to submit these findings as a separate article. Thus, in this article, we further added a brief description of the fin-ray phenotype as follows.

In the mutant fish, we observed segmentation in the soft rays, although the spacing between segments was uncertain. Restricted fin-ray growth may also contribute to the shortening of individual soft rays. The curled phenotype was not consistently observed among mutant individuals, and appears to result from the posterior finfold margin failing to extend properly.

To more clearly describe these phenotypes, we have revised Figure 2 by adding yellow arrowheads indicating examples of soft-ray segmentation in the knockout fish. In addition, we have added explanatory text to the manuscript and included new extended figures (Extended Data Fig. 4) showing mutant individuals without the curled caudal-fin phenotype.

Line 164-167, “We observed that the intervals of the segments of the soft rays were uncertain (yellow arrowheads in Fig. 2c and Extended Data Fig. 2). The restriction of fin ray growth may result in the shortening of each soft ray. The curled phenotypes were not always observed in the mutant fish, with this phenomenon being caused by the edge of the caudal part of the fin fold failing to grow.”

Extended Data Figure 4 Actinotrichia distribution and spiny- and soft-ray bone morphology in the *actinodin1^{-/-}/actinodin2^{+/-}* knockout *Melanotaenia praecox*.

White arrowheads indicate abnormal bending of the soft rays in the knockout fish. Yellow arrowheads show aberrantly positioned segmental structures in the soft rays of the knockout fish. Actinotrichia were labeled with DAFFM DA (green), and the spiny- and soft-ray bones were labeled with alizarin red (magenta).

Reviewer comment 1-4:

Regarding Figure 5b, a previous study by Hoch et al. (PNAS 2021) demonstrated that DMH1 treatment causes spiny rays in cichlid fish to transform into soft rays. Perhaps the DMH1 treatments were carried out after the initial development of the spiny ray, but that should be properly described. In this study, larvae were incubated for 10 days, which is a considerable duration. Can the authors rule out the possibility that DMH1 treatments caused the spiny ray to develop a soft ray-like structure, resulting in the observed deformation? Do DMH1-treated spiny rays retain the properties of spiny rays?

Response 1-4:

This is an important point, and we thank the reviewer for raising it. We carefully considered the possibility that DMH1 treatment caused the spiny rays to acquire soft ray-like characteristics. However, several lines of evidence suggest that this is unlikely.

First, in the previous study by Hoch et al. (2021), cichlid fish were treated with DMH1 *before* the appearance of fin rays. In their experiment, inhibition of BMP signaling altered the future fate of fin mesenchymal cells. In contrast, in our study, we treated *M. praecox* individuals *after* the spiny and soft rays had already formed, thereby perturbing fin mesenchymal cells only after osteoblast differentiation had occurred. Thus, the timing of BMP inhibition differs markedly between the two studies.

Furthermore, if DMH1 treatment had induced a transition in osteoblast identity from a spiny-ray type to a soft-ray type, we would expect the transformed spiny rays to exhibit certain features characteristic of soft rays. For example, we would have detected actinotrichia-like collagenous structures or segmentation within the transformed spiny rays. However, our analyses confirmed that none of these features appeared in DMH1-treated spiny rays.

While we cannot completely exclude the possibility of subtle changes in osteoblast identity, our results do not support the hypothesis that DMH1 treatment induced a major transformation from spiny rays to soft rays. We have revised the manuscript to incorporate this discussion.

Line 336-339, “Despite BMP signaling inhibition by DMH1, the spiny rays retained their characteristic unsegmented and rigid morphology. No actinotrichia-like collagenous structures or segmentation were observed at the tips of the spiny rays in treated specimens (Fig. 5b).”

Line 425-434, “Hoch et al. (2021)¹¹ reported that inhibition of BMP signaling prior to fin-ray formation induced homeotic transformations of prospective spiny rays into soft rays. In contrast, our DMH1 treatment, applied after the spiny rays had already formed, did not result in any apparent transformation of spiny rays into soft ray-like structures. Although we cannot entirely rule out minor shifts in osteoblast identity, the absence of segmentation or actinotrichia-like collagenous structures in treated specimens indicates that BMP inhibition did not induce a major transformation from spiny to soft rays once the spiny rays had formed. Taken together, these findings suggest that BMP signaling primarily contributes to establishing the anterior–posterior division of the dorsal and anal fins into spiny- and soft-ray domains, but may not drive a qualitative switch in osteoblast identity after cell differentiation has occurred.”

Reviewer comment 1-5:

Additionally, I believe the study by Hoch et al. is highly relevant to this paper and influences the authors' work. While their paper is cited in this manuscript (11), a more detailed description is needed. Their findings, which illustrate the establishment of spiny and soft-rayed fin domains in acanthomorph fishes, provide essential molecular insights into the differences between spiny and soft-rayed fins.

Response 1-5:

We agree that the study by Hoch et al. (2021) provides important molecular insights into the distinction between spiny- and soft-rayed fins. In the revised manuscript, we have expanded our description of this study to clarify how their findings relate to our work, as shown below:

Line 416-424, “A previous study reported that distinct transcription factor signatures characterize the spiny- and soft-ray domains¹¹. The spiny-ray domain specifically expresses *alx4a/b* and *tbx2b*, whereas the soft-ray domain expresses *hoxa13a/b*. Our study revealed differences in developmental mechanisms between spiny and soft rays at both cellular and ECM levels. These differences may be regulated by transcription factors that exhibit mutually exclusive expression patterns in the two domains. Consistent with this idea, ablation of *hoxa13a/hoxd13a*-expressing fin fold mesenchymal cells reduced the formation of actinotrichia. Taking these findings together with their expression specificities, *alx4a/b* and *tbx2b* may promote the mesenchymal cell aggregation characteristics of spiny rays, whereas *hoxa13a/b* may enhance the actinotrichia synthesis essential for soft rays.”

Reviewer comment 1-6:

*In Figure 6, the Runx2/pSmad signals largely overlap in the regions surrounding the tips of the laterally protruding spiny ray in filefish. In contrast, Figures 5a and 5b, which utilize *Melanotaenia praecox*, show less overlap in the Runx2/pSmad signals, particularly at the tips of the spiny fins. Could this discrepancy be attributed to the presentation of optical sections? The images should be updated to display the z-stack images to clearly show the distributions of the pSmad signals.*

Response 1-6:

We sincerely apologize for our incomplete explanation, which led to this confusion. You have raised an important point. The colocalization patterns do indeed appear to be different between the two species.

Regarding the technical suggestion to use z-stack images, we apologize for not clarifying our methodology. All images in Figure 5a and the right-hand three columns of Figure 6c are optical section images. For the *M. praecox* samples, optical sections were captured using a pinhole size of 200 μm , whereas for the *S. cirrhifer* samples the pinhole was adjusted to 202.1 μm . In confocal microscopy, the pinhole is the key component that gives the system its ability to produce thin optical sections. The difference in pinhole size between the images of the two species is minimal, and it does not affect the degree of overlap between Runx2 and pSmad signals.

We intentionally presented optical sections because they are the most rigorous and accurate method for assessing true colocalization within individual cells. As you no doubt know, z-stack imaging can create false signal overlaps from cells at different depths because of the collapsing of the 3D data. This would make it more difficult to determine the true colocalization status of Runx2 and pSmad within individual cells. Therefore, we believe the optical sections that we present are the most scientifically accurate data for this analysis.

As the reviewer pointed out, the colocalization patterns of Runx2 and pSmad differ between the lateral protrusions of the dorsal spine in *S. cirrhifer* and the tips of the spiny rays in *M. praecox*. We are very grateful to you for highlighting this important finding. We interpret these differences as reflecting genuine variation in BMP signaling intensity, which may in turn contribute to the distinct morphogenetic processes in these structures. To make this biological difference clearer (as requested), we have added arrowheads in Figures 5 and 6. We have also added this important comparative point to the Discussion, because we agree that the differences reflect genuine variation in BMP signaling intensity, which may in turn contribute to the distinct morphogenetic processes. We have revised the manuscript to include this explanation.

Line 462-467, “Our pharmacological inhibition assay in *M. praecox* demonstrated that BMP signaling in spiny rays regulates the thickness of the accumulating bony matrix (Fig. 5b). In addition, the colocalization patterns of Runx2 and pSmad1/5/9 differed between the spiny rays of *M. praecox* and the thorny spine of *S. cirrhifer*. Together, these data suggest that in the thorny spine of *S. cirrhifer*, position-dependent modifications in BMP signaling intensity may regulate thorn formation.”

Reviewer comment 1-7:

*How many larvae of *Stephanolepis cirrhifer* were examined using immunostaining? The staining results from other individuals should be presented as supplementary data to confirm reproducibility.*

Response 1-7:

We sincerely apologize for our oversight regarding the sample size of the immunostaining data using *Stephanolepis cirrhifer* larvae. We have added the sample size at Line 356 and immunostaining images from other individuals in Extended Data Figure 7.

Extended data figure 7. Premature osteoblasts with BMP signaling are spatially modified from the tip to the lateral protrusions of the first dorsal-fin spiny ray in *Stephanolepis cirrhifer*.

a, b Distributions of Runx2-positive cells (green) and pSmad1/5/9-positive cells (magenta) in the dorsal spine of two *S. cirrhifer* specimens, stained by immunohistochemistry with DAPI (white). Yellow arrow heads show examples of pSmad1/5/9 and Runx2-positive cells. Dorsal-spine bones are outlined in orange. The left-most picture is a confocal Z-stack image, and the others are optical sections obtained by confocal microscopy.

Reviewer comment 1-8:

Regarding the evolution of spiny rays, do the authors think they evolved from soft rays by losing actinotrichia, or were they acquired independently?

Response 1-8:

Thank you very much for this comment. To answer this question, we have added text to the Discussion, as below.

Line 435-442, “Building on previous studies, our findings allow us to speculate on the evolutionary origin of spiny rays. One simple hypothesis is that spiny rays evolved from soft rays through the loss of their actinotrichia in the anterior part of each fin. However, the soft-ray morphology of the *and1*^{-/-}; *and2*^{+/-} mutants did not resemble that of spiny rays because the soft rays lacked their straight morphology. Therefore, this hypothesis is not supported by the data. To definitively test the evolutionary hypothesis regarding the evolutionary acquisition of spiny rays, further studies are required on developmental mechanisms in species belonging to lineages of the teleost phylogeny that diverged earlier than the Acanthomorpha did.”

(Minor points)

Reviewer comment 1-9:

In Fig. 1a, spiny and soft rays stained with Alizarin red are shown. Please specify which fin rays belong to which fish.

Response 1-9:

As suggested, we have added information about the fin-ray samples stained with alizarin red in the legend of Figure 1, as below.

Line 111-112, “The spiny- and soft-ray samples stained with alizarin red are the dorsal fins of *Melanotaenia praecox*.”

Reviewer comment 1-10:

Lines 530-532: Was 5 µl injected into each embryo? That seems like a large volume; please confirm.

Response 1-10:

Thank you for pointing this out. We have corrected the error below:

Line 648-650, “We prepared a 5-µl solution containing 100 ng of each sgRNA, 1,250 ng of Cas9 protein, and 0.5 µl of phenol red in nuclease-free water, and then 2–3 nL of this solution were injected into the single cell of the embryo.”

Reviewer comment 1-11:

Figure 5b: Are the green signals on several dots non-specific? Please clarify this point.

Response 1-11:

We apologize for being unclear on this part. We added an explanation of these dots as below:

Line 329-332, “In the immunohistochemistry images for pSmad1/5/9 (leftmost panels of Fig. 5b), several punctate signals are visible in both control and DMH1-treated fish. In previous studies using the same antibody, pSmad1/5/9 immunoreactivity was prominent in nuclei^{58,59}. Therefore, the punctate signals observed in our images may represent non-specific staining.”

Reviewer comment 1-12:

Line 210: While both DAFFM and DAR4M are claimed to detect actinotrichia, I am unclear about the differences between them.

Response 1-12:

We apologize for the poor explanation. We added a brief explanation of the difference of these two small molecules as below.

Line 244-247, “DAR-4M is a small fluorescent molecule similar to DAF-FM that is widely used for NO detection; however, the two molecules exhibit different fluorescence emission wavelengths^{32,45}. Recent studies have shown that DAR-4M can bind to collagenous structures such as actinotrichia^{32,47}.”

Reviewer #2 (Remarks to the Author):

Reviewer comment 2-1:

*The manuscript entitled “Actinotrichia-independent developmental mechanisms of spiny rays facilitate the morphological diversification of Acanthomorpha fish fin”, submitted to Nature Communications, investigates developmental differences between spiny rays and soft rays in rainbowfish (*Melanotaenia preacox*), with additional comparative studies from the highly modified spiny rays of the filefish (*Stephanolepis cirrhifer*).*

To address their objectives, the authors employ a comprehensive suite of techniques, including micro-CT, electron microscopy, immunofluorescence, and in-situ hybridization, to demonstrate key distinctions in fin ray development and anatomy at different scales (from the entire spine to the cellular level). Specifically, they show that spiny rays are formed by an accumulation of mesenchymal cells where BMP signaling plays an important role, which differs from soft rays that rely on actinotrichia as a structural scaffold. Furthermore, the authors identify similar spiny-ray developmental mechanisms in the filefish, suggesting that this actinotrichia-independent mode of growth may have facilitated the morphological diversification of fin rays within Acanthomorpha.

Overall, the manuscript is well-executed and clearly written, with well-organized flow of ideas. The experimental design is robust to address the question proposed (although see general comments about sample size and sexes used), and the integration of the multiple methods provides a holistic and convincing analysis. The figures clearly show the results and are easy to follow. The research question is original, particularly because little is known about the skeletal development of acanthomorphs. Therefore, this research is of interests across multiple fields, including fish biology, physiology, skeletal development, and evolution. However, I suggest the authors to revise my general comments regarding this topic.

My main concern is the lack of discussion and consideration of the bone type the authors are studying (acellular bone) as well as comparing to (cellular bone), which I consider that is crucial to understand the results and do an accurate comparative analysis. I hope my review will help further improve the clarity and impact of this work.

Response 2-1:

We fully appreciate these comments. We have made revisions to the manuscript in line with your comments and suggestions.

General comments

Reviewer comment 2-2:

Acellular bone skeletogenesis.

One aspect that stood out to me is that the authors do not mention, or consider, the type of bone that forms the skeleton of both species of fishes that are studied here. The majority of Neoteleostei, and thus acanthomorphs (except some species like tunas) have acellular bone (i.e. a bone without osteocytes). This type of bone is also found in some Euteleostei. In contrast, more basal groups of fishes are characterized by cellular bone (i.e. a bone with osteocytes), including Ostariophysii where cypriniformes (zebrafishes) and siluriformes (catfishes) are part of.

This difference in bone type could be one of the reasons why the development of spiny rays in zebrafish and catfish (species that authors use to compare their results) may differ from that in the species studied

here. Therefore, the spiny ray development described in the manuscript may probably be related with the periosteal osteogenesis of acellular bone. Check also the anatomical terminology used (e.g. basement membranes), as some of the structures may have been already described in Moss (1961). This point can be a very interesting topic to include and discuss, since the formation of acellular bone has not been largely studied.

For these reasons, I suggest that the authors consider and discuss the acellular bone formation in the manuscript, as well as to be cautious in describing this developmental process as “unique” to Acanthomorph spiny rays, as what the authors may be describing reflect general features of acellular bone skeletogenesis rather than a distinct or novel pathway of spiny rays. Additionally, the importance of BMP signaling, while certainly relevant, is not surprising as this superfamily of proteins is well known to have a crucial role during bone development across vertebrates.

Below I included several papers that discuss the differences between acellular or cellular bone that the authors may find useful.

References

- Davesne, D., Meunier, F. J., Schmitt, A. D., Friedman, M., Otero, O., & Benson, R. B. (2019). The phylogenetic origin and evolution of acellular bone in teleost fishes: insights into osteocyte function in bone metabolism. *Biological Reviews*, 94(4), 1338-1363.

- Shahar, R., & Dean, M. N. (2013). The enigmas of bone without osteocytes. *BoneKEy reports*, 2, 343.

- Moss, M. L. (1961). Osteogenesis of acellular teleost fish bone. *American Journal of Anatomy*, 108(1), 99-109.

- Weiss, R. E., & Watabe, N. (1979). Studies on the biology of fish bone. III. Ultrastructure of osteogenesis and resorption in osteocytic (cellular) and anosteocytic (acellular) bones. *Calcified tissue international*, 28(1), 43-56.

- Moss, M. L. (1963). The biology of acellular teleost bone. *Annals of the New York Academy of Sciences*, 109(1), 337-350.

- Dean, M. N., & Shahar, R. (2012). The structure-mechanics relationship and the response to load of the acellular bone of neoteleost fish: a review. *Journal of Applied Ichthyology*, 28(3), 320-329.

Response 2-2:

We thank you for raising this very interesting and crucial point. Your comment has prompted us to clarify our argument regarding the evolution of spiny rays and the context of acellular bone.

The acquisition of spiny rays in Cypriniformes and Siluriformes is likely independent of that in Acanthomorpha. Hoch et al. (2021) showed that the early developmental phase of spiny rays in African catfish resembles that of typical soft rays. In addition, Kubicek et al. (2019) demonstrated that soft rays possess actinotrichia at their distal tips. These observations suggest that the spiny rays found in non-acanthomorph fishes are homologous to soft rays, rather than to the acanthomorph spiny rays. For this reason, we consider that the differences in spiny-ray morphology between acanthomorph and non-acanthomorph fishes are primarily caused by differences in osteoblast distribution and physical constraints associated with the presence or absence of actinotrichia.

We agree with your point about acellular bone being a critical distinction. However, we propose that the acellular bone type alone cannot explain the developmental differences we observed, with this being

based on two key lines of evidence. First, the phylogenetic timing of the bone type transition is according to the emergence of the Euteleostei, and this shows a mismatch with the timing of the emergence of spiny rays. Second, our study's primary comparison is between spiny rays and soft rays within the same species. In these fish, both structures are acellular, meaning the bone type itself cannot be the variable that explains the difference in their developmental mechanisms (i.e., actinotrichia-dependence vs. independence).

We agree with your hypothesis that the evolutionary acquisition of acellular bone may have facilitated the subsequent emergence of acanthomorph spiny rays. This is an interesting point that enriches our discussion. Following your suggestion, we have now added this important speculation to the manuscript, as shown below:

Line 442-449, “Furthermore, the evolution of bone types might have contributed to the acquisition of certain features of spiny rays. The majority of Euteleostei, including Acanthomorpha (except for some species such as tunas), possess acellular bone, which lacks osteocytes⁶⁰. In contrast, more basal groups of fishes are characterized by cellular bone, which contains osteocytes⁶⁰. Although the phylogenetic timing of the transition from cellular to acellular bone does not coincide with the emergence of spiny rays, the prior establishment of acellular bone may have facilitated the acquisition of features such as the fusion of the two hemirays and the stiff structure of spiny rays.”

In light of your comments, we have carefully reviewed the anatomical terminology in our manuscript. We appreciate this prompt to ensure precision, and although we have confirmed our existing terminology is appropriate, we are grateful for the advice.

We agree that “unique” was a strong term because some aspects of spiny-ray osteogenesis may be shared with other types of bones. We have removed this word and revised the sentence as follows:

Line 87-88, “Based on our observations, we propose that the developmental mechanisms of spiny rays may have facilitated their morphological diversification.”

Reviewer comment 2-3:

Sample size.

*The authors specify sizes in each section of the Results, but not in Methods. I recommend also including a detailed breakdown in the Methods section, indicating the number of animals used at each developmental stage and for each technique (e.g. How many *S. cirrhifer* and *M. praecox* where μ CT-scanned?, this is not specifically clarified anywhere). Additionally, in some instances, it is unclear which ontogenetic stage is used for a given specific technique (larvae, juveniles, adults?). Lastly, in some cases it is unknown whether the reported sample size in Results refers to the number of rays examined or the number of individual animals. Since multiple rays can be studied from a single specimen, it would be helpful to clarify the number of animals studied similar to how it is done in L130.*

Response 2-3:

Thank you for this important comment. We apologize for the lack of detail in the Methods section. To address these concerns, we have now included a detailed breakdown of the number of individuals used for each technique in the Methods section. Furthermore, we have clarified that all reported sample sizes (n) refer to the number of individuals, and we now explicitly describe the developmental stages of the samples in both the Results and Methods sections.

In the original manuscript, we used one individual each of *S. cirrhifer* and *M. praecox* for the μ CT scanning. To increase the robustness of our data, we have performed additional μ CT scans on both *S. cirrhifer* and *M. praecox* individuals. The corresponding data and additional methodological details have been added to the manuscript. Additionally, we have made the CT data publicly available on figshare.

Reviewer comment 2-4:

Animal sex.

The authors do not specify whether the specimens used were males or females. Several teleosts, including M. praecox, are sexual dimorphic, with males exhibiting enlarged caudal spines.

Response 2-4:

You have raised important points. For *M. praecox*, we did not distinguish the sex for most experiments (only for μ CT scanning of adults) because the sex of larval and juvenile *M. praecox* cannot be distinguished; the sex determination mechanisms remain unclear for this species. Consequently, it is not currently feasible to distinguish the sex of our experimental samples using PCR-based methods or other investigations. To address the potential issue you have raised, we conducted an additional observation on young adult *M. praecox* (Extended Data Figure 8), whose sexes are distinguishable by their body coloration. We found no significant differences in fin bone morphologies between males and females of this stage. This result indicates that skeletal sexual dimorphism in this species likely appears after sexual maturation. Therefore, we concluded that it was not necessary to distinguish the sex of the larval and juvenile specimens used in our experiments.

Regarding *S. cirrhifer*, although sex chromosome system and sexual dimorphisms in the adult second dorsal fin have been reported in Murofushi et al. (1980) and Fujita (1955), we did not distinguish the sex of our specimens. According to Kwon et al. (2021), sexual maturation in this species occurs at a total length of approximately 11.7 cm. The specimens used for experiments (0.9–1.4 cm Standard Length) were all substantially smaller than this size, and thus would not be expected to have developed sexual dimorphisms. Furthermore, sexual dimorphisms of the dorsal spine have not been reported. Taken together, we concluded that distinguishing the sex of our *S. cirrhifer* specimens was unnecessary.

To clarify our rationale for not distinguishing sexes in the manuscript, we have added the aforementioned explanations to the Methods section and new supporting data as an extended figure (Extended Figure 5).

Line 523-529, “When observing the dorsal fin bone morphologies of young male and female fish (male, $n = 3$; female, $n = 3$), whose sexes are distinguishable by their body coloration³⁶, we did not find any significant differences in the fin bone morphologies between males and females at this stage (Extended Data Fig. 5). Furthermore, we did not have a reliable technique to distinguish the sex of larval and juvenile *M. praecox* using PCR-based methods or other investigations, and thus did not distinguish the sex of the larval and juvenile specimens in our experiments.”

Line 531-539: “Although sex chromosome system and sexual dimorphisms in the adult second dorsal fin have been reported in this species^{66,67}, we did not distinguish the sex of our specimens because a reliable PCR-based genotyping method has not yet been established. According to Kwon et al. (2021)⁶⁸, sexual maturation in this species occurs at a total length of approximately 11.7 cm. The specimens used in our experiments were all significantly smaller than this size and would thus not have been expected to have developed sexual dimorphisms. Furthermore, sexual dimorphisms of the dorsal spine have not been reported. Taken together, we concluded that distinguishing the sex of our *S. cirrhifer* specimens was unnecessary.”

Extended Data Figure 8. Fin bone morphology in the dorsal fin of male and female *Melanotaenia praecox*.

Calcein staining in young adult *M. praecox* showing bone morphology in the dorsal fins. Yellow asterisks indicate the tungsten needles that were used to spread their dorsal fins.

Specific comments:

Abstract

Reviewer comment 2-5:

- L22. The hyphenation of “spiny rays” and “soft rays” varies throughout the manuscript. In some instances, the terms are hyphenated (e.g., lines 23, 24, 27), while in others they are not (e.g., lines 22, 48, 49, 56). Please check for consistency and revise accordingly.

Response 2-5:

Thank you for pointing out the inconsistency in the hyphenation of “spiny ray” and “soft ray.”

We have carefully revised the manuscript to ensure consistency. Specifically, we use “spiny-ray” and “soft-ray” when these terms modify another noun (e.g., *spiny-ray development*), and “spiny ray” and “soft ray” otherwise (e.g., *the spiny rays*). This usage conforms to standard English grammar conventions for compound adjectives.

Reviewer comment 2-6:

- L22. “their bone structure” is a vague descriptor in this context. What about “their supporting skeletal structure”.

Response 2-6:

We revised the manuscript according to your suggestions.

Reviewer comment 2-7:

- L30. “the lateral protrusions equip the BMP positive osteoblast condensation...”. The verb “equip” here is a bit confusing. What about: “the lateral protrusions are associated with BMP-positive osteoblast condensation”

Response 2-7:

We revised the manuscript according to your suggestions.

Reviewer comment 2-8:

- L36. "... ECM usage would be major factors driving the morphological diversification in animals". I recommend focusing this sentence specifically on acanthomorphs, rather than using the broader term "animals".

Response 2-8:

We apologize for the insufficient explanation. We hypothesize that the evolutionary mechanisms underlying diversification of the acanthomorph spiny rays may represent a general process occurring in various cases of animal evolution. Thus, we suggest that cell distribution and ECM usage could be regarded as factors driving morphological diversification not only in Acanthomorpha, but also across a broad range of animal taxa. However, this hypothesis has not yet been tested, and direct evidence supporting this possibility is still lacking. We have revised the manuscript to take these points into account.

Line 37-39, "This suggests that variation in cell distribution and ECM usage may be important contributors to morphological diversification, not only in Acanthomorpha, but also in other animal taxa."

To clarify our suggestion, we have also revised text in the Discussion section.

Line 501-505, "We suggest that, in addition to genome duplications⁶¹, acquisition of regulatory cis-elements⁶², and the innovation of cell types⁶³ and molecular interactions⁶⁴, variation in cell distribution and the usage of ECM, such as in the actinotrichia in our fish models, may also be major factors determining the extent of morphological diversification in a broad range of animal evolution."

Introduction

Reviewer comment 2-9:

- L48. "...and mostly branching,...". Maybe replace "mostly branching" to "typically branched".

Response 2-9:

We revised the manuscript according to your suggestions.

Reviewer comment 2-10:

- L54 – 55. Are the sucking disc of remoras or the illicium considered "spiny rays"? or they have been so modified that they are no longer considered as such?

Response 2-10:

We apologize for the unclear explanation. The sucking discs of remoras and the illicium are considered to be homologous structures to spiny rays. We refrain from proposing a single definitive name here because various naming schemes are possible and each emphasizes different aspects. Consistent with our original manuscript, we avoid making a determination about whether these structures should be regarded as spiny rays.

Reviewer comment 2-11:

- L72. "...there anal fin." There is a typo here, it should be "their anal fin".

Response 2-11:

As suggested, we corrected this error.

Reviewer comment 2-12:

- L78. "...with that of soft-ray development." Those of soft-ray development instead, to be consistent with the plural subject ("matrix and cell dynamics").

Response 2-12:

As suggested, we corrected this error.

Results

Reviewer comment 2-13:

- L92 – 94. This sentence reads awkwardly and may be either missing some information or be poorly structured. I recommend revising it for clarity, as its current form makes the intended meaning difficult to follow.

Response 2-13:

We appreciate your suggestion. The sentence has been revised for clarity and readability. The revised version now reads as follows:

Line93-94, "To identify differences in morphogenesis between soft and spiny rays, we first visualized their developmental changes using two different staining methods¹²."

Reviewer comment 2-14:

- L114, Figure 1. It is "yellow" instead of "white".

Response 2-14:

As suggested, we corrected this error.

Reviewer comment 2-15:

- L128. "DAFFM is a small fluorescent molecule that broadly used for..". Instead: "DAFFM is a small fluorescent molecule that has been broadly used for...".

Response 2-15:

We revised the manuscript according to your suggestions.

Reviewer comment 2-16:

- L131. Delete "obviously".

Response 2-16:

We revised the manuscript according to your suggestions.

Reviewer comment 2-17:

- L132. Regarding the statement “fine actinotrichia were present at the future position of spiny rays in the median fin fold”. This statement is not clearly supported by the figure. In the image (Fig 2a), the actinotrichia do not appear to be specifically localized at the future position of the spiny rays. In fact, it doesn't look like they are found in a specific position. I recommend clarifying the figure or revising the interpretation in the text to better reflect what is shown.

Response 2-17:

We appreciate your helpful comment. We agree that the actinotrichia are not specifically localized at the future positions of spiny rays. Accordingly, we have revised the sentence to clarify the description as follows:

Line134-136, “Before the emergence of spiny rays (i.e., in DF-st3 larvae), relatively small and less developed actinotrichia, compared with those at the tips of the soft rays, were observed in the median fin fold, including regions that later give rise to the spiny rays.”

Reviewer comment 2-18:

- L135 – 136. In the spiny and soft rays, the DAFFM only stains actinotrichia? If not, which molecules form the brightest bulk of the rays? See image below:

Response 2-18:

We apologize for the unclear explanation. DAFFM staining visualizes collagenous components, including both actinotrichia, bone collagen, and tendon of the segments in the soft rays. We have clarified this point in the added text as follows:

Line139-140, “DAFFM staining visualizes collagenous structures, including actinotrichia, bone collagen, and tendon of the segments in the soft rays^{32,45}.”

Reviewer comment 2-19:

- L209: “n=3”. The number of spiny tails (sample size) sampled are from 3 different animals?

Response 2-19:

We revised the explanation about sample sizes as you suggested in *Reviewer comment 2-3*.

Line242-244, “First, we examined the 3D cell distribution at the tips of spiny rays (DF-st4 larvae, n = 3) and soft rays (DF-st4 larvae, n = 3) by staining the nucleus and collagen structures simultaneously.”

Reviewer comment 2-20:

- L213: “...such condensed mesenchyme cells”. How the authors know that these are mesenchyme cells if they did not use any markers, only the staining of the nuclei? If the authors are defining mesenchyme cells based on their rounder, larger morphology, they could be also osteoblasts or other precursor cell?

Response 2-20:

We apologize for the mistake. In this context, our intention was to indicate that the non-epithelial cells condensed at the tip of the spiny rays, but not at the tip of the soft rays. Therefore, we have replaced the word “mesenchyme” with “mesenchymal cells” in the relevant sentence.

Reviewer comment 2-21:

- Figure 3: There is a typo in SYTO13 (nucleus).

Response 2-21:

As suggested, we corrected this error.

Discussion

Reviewer comment 2-22:

- L334: using the “unique” adjective here is unnecessary. The authors do not know if this mechanism is unique to spiny rays or to acanthomorphs, since this process can be happening in other skeletal elements or in other clades (perhaps is how the acellular bone is formed?).

Response 2-22:

Thank you for this suggestion. We omitted the “unique” adjective and revised the sentence as follows:

Line391-392, “Our study confirms that some features of the morphogenetic mechanisms underlying spiny-ray development in acanthomorph fish differ from those of soft-ray development.”

Reviewer comment 2-23:

- L344: Secondly?

Response 2-23:

We apologize for the insufficient explanation. We used the word “secondarily” to indicate that the developmental mechanisms occurring at the tips of spiny rays were also utilized at the lateral protrusions of the dorsal spine in *S. cirrhifer*. To avoid confusion, we have removed this term and revised the manuscript as follows.

Line 373-374, “Figure 6. Premature osteoblasts with BMP signaling are spatially modified from the tip to the lateral protrusions of the first dorsal-fin spiny ray in *Stephanolepis cirrhifer*.”

Line 400-402, “The acquisition of the thorny spine in *S. cirrhifer* may have resulted from a spatial modification of an ancestral developmental mechanism that was originally established at the tips of spiny rays.”

Line 494-496, “The pre-established independence from actinotrichia and the autonomous osteogenic mechanisms at the tips of spiny rays may have been spatially modified and redeployed at alternative positions.”

Reviewer comment 2-24:

- L361 – L363: *The fact that the developmental process that forms the spiny rays described in this manuscript is different of the process of the formation of spiny rays in Siluriformes may be because catfishes have cellular bone (see my comments above).*

Response 2-24:

Thank you for bringing this to our attention. We have carefully considered this point, and our detailed explanation can be found in Response 2-2.

In brief, recent studies (e.g., Hoch et al., 2021) suggest that the “spiny rays” in Siluriformes (catfish) are developmentally homologous to soft rays (i.e., they are actinotrichia-dependent). As this point is integral to the main discussion on bone types, our full and detailed explanation can be found in Response 2-2.

Reviewer comment 2-25:

- L406 – L411: *this text looks like doesn't belong here?*

Response 2-25:

The text in lines L406–L411 is correctly placed. We apologize for the confusion caused by a simple formatting error that resulted in this section being displayed in a smaller font size.

Methods

Reviewer comment 2-26:

- *Add sample sizes for each technique (see my comment above).*

Response 2-26:

As suggested, We revised the manuscript according to your suggestions.

Reviewer comment 2-27:

- L445: *after fixation, where the samples directly transferred to 70% ETOH? Or the authors dehydrated them using increasing concentration steps?*

Response 2-27:

We apologize for our unclear explanation. We revised the methodology as follows:

Line 554-557, “Samples of *M. praecox* adults were fixed with 10% formaldehyde at room temperature overnight, followed by dehydration using an ethanol series (50% EtOH, 70% EtOH, 80% EtOH, 90% EtOH, 95%EtOH, 100% EtOH, 95% EtOH, 90%EtOH, 80%EtOH, 70%EtOH). These samples were then stored in 70% ethanol.”

Reference

- Höch, R., Schneider, R. F., Kickuth, A., Meyer, A., & Woltering, J. M. (2021). Spiny and soft-rayed fin domains in acanthomorph fish are established through a BMP-gremlin-shh signaling network. *Proceedings of the National Academy of Sciences of the United States of America*, 118(29).

- Kubicek, K. M., Britz, R., & Conway, K. W. (2019). Ontogeny of the catfish pectoral-fin spine (Teleostei: Siluriformes). *Journal of Morphology*, 280(3), 339–359.
- Murofushi, M., Oikawa, S. Nishikawa, S. Yoshida, T. H. (1980) Cytogenetical Studies on Fishes, III. Multiple Sex Chromosome Mechanism in the Filefish, *Stephanolepis cirrhifer*, The Japanese Journal of Genetics, 1980, Vol. 55, No. 2, 127-131,
- Fujita, S. (1955) On the development and prelarval stages of the file-fish, *Monacanthus cirrhifer* Temminck et Schlegel. *Sci. Bull. Fac. Agr., Kyushu Univ.*, 15: 229-234. (In Japanese with English summary.)
- Kwon, H. C., Lee, J. B., Zhang, C. I., Lee, D. W., Choi, Y. M. (2011) Maturation and Spawning of Filefish, *Stephanolepis cirrhifer* in the East Sea of Korea. *KOREAN JOURNAL OF ICHTHYOLOGY*, Vol. 23, No. 2, 111-118.

We thank you very much for reviewing our manuscript. We have revised it following careful examination of your suggestions. Your comments helped us to greatly improve the quality of our paper, and we express our gratitude to both of you for your constructive suggestions.

Reviewer #2 (Remarks to the Author):

Reviewer comment 2-1:

This is the second time that I have reviewed the manuscript entitled “Actinotrichia-independent developmental mechanisms of spiny rays facilitate the morphological diversification of Acanthomorpha fish fins”, submitted to Nature Communications.

The authors have adequately addressed my previous comments. In particular, they have responded to my suggestions regarding the discussion of cellular versus acellular bone, and they have clarified the sex of the specimens used whenever possible.

Overall, I think the authors present a robust and carefully conducted study. This work offers valuable insights into the developmental mechanisms of spiny and soft rays which, in case of spiny rays, likely contribute to the highly diversified morphologies observed in some acanthomorph species.

I have noted a few minor grammatical errors throughout the manuscript, but aside from these minor issues, the manuscript reads well and is scientifically sound.

Response 2-1:

We thank the reviewer for their positive evaluation of our revised manuscript and for taking the time to re-review it. We are pleased that our responses to the previous comments were found to be adequate.

We have carefully corrected the minor grammatical errors noted by the reviewer. We sincerely appreciate the reviewer’s constructive feedback and support.

Reviewer comment 2-2:

- L50: “Branched” instead of “branching”

Response 2-2:

As suggested, we corrected this error.

Reviewer comment 2-3:

- L248: “We found that mesenchymal cells condensed at the tips of spiny rays, whereas such condensed mesenchymal cells were not detected at the tips of soft rays (Fig. 3a)”. Personally, I don’t see the condensation of mesenchymal cells in Figure 3a.

Response 2-3:

We appreciate your helpful comment. Accordingly, we have revised the sentence to clarify the description as follows:

Line247-249, “We found that mesenchymal cells are multilayered at the tips of spiny rays (Fig. 3a-left), whereas such condensed mesenchymal cells were not detected at the tips of soft rays (Fig. 3a-right).”

Reviewer comment 2-4:

- L254: “Zebrafish” instead of “Zebrafish”

Response 2-4:

As suggested, we corrected this error.

Reviewer comment 2-5:

- L280: “a Cell, ECM, and bone condition”. The ECM is not particularly visible in a.

Response 2-5:

We apologize for the unclear explanation. DAR4M staining visualizes only collagenous components. Then, we have clarified this point in the added text as follows:

Line280, “a Cell, collagen, and bone conditions of the dorsal fin in a larva at DFst-4.”

Reviewer comment 2-6:

- Small yellow and pink numbers, present in figure 3 and Extended Data figure 6, that indicate regions located at the top-left corner. These are very difficult to see, maybe adding a black background may help.

Response 2-6:

We revised the figures according to your suggestions.

Figure3

Extended Figure6

Reviewer comment 2-7:

- L360: "Larva" is singular, therefore it should be "has".

Response 2-7:

As suggested, we corrected this error.

Reviewer comment 2-8:

- L411: "Alternative to" instead of "Alternative of"

Response 2-8:

As suggested, we corrected this error.